# Roughness of Fracture Surfaces in Numerical Models and Laboratory Experiments

Steffen Abe[1] and Hagen Deckert[1]

[1]Institute for geothermal resource management, Berlinstr. 107a, 55411 Bingen, Germany

**Correspondence:** Steffen Abe (s.abe@igem-energie.de)

**Abstract.** We investigate the influence of stress conditions during fracture formation on the geometry and roughness of fracture surfaces. Rough fracture surfaces have been generated in numerical simulations of triaxial deformation experiments using the Discrete Element Method and in a small number of laboratory experiments on limestone and sandstone samples. Digital surface models of the rock samples fractured in the laboratory experiments were produced using high resolution photogrammetry. The roughness of the surfaces was analyzed in terms of absolute roughness measures such as an estimated joint roughness coefficient (JRC) and in terms of its scaling properties. The results show that all analyzed surfaces are self-affine, but with different Hurst exponents between the numerical models and the real rock samples. Results from numerical simulations using a wide range of stress conditions to generate the fracture surfaces show a weak decrease of the Hurst exponents with increasing confining stress and a larger absolute roughness for transversely isotropic stress conditions compared to true triaxial conditions. Other than that, our results suggest that stress conditions have little influence on the surface roughness of newly formed fractures.

## 1 Introduction

It is well known that surfaces of faults and fractures in rocks are rough at all scales (Brown and Scholz, 1985; Hobbs, 1993; Power and Durham, 1997; Candela et al., 2012). The roughness of fracture surfaces is important for a range of geological processes such as the mechanical behavior of faults (Okubo and Dietrich, 1984; Griffith et al., 2010; Candela et al., 2011a, b; Angheluta et al., 2011; Ahmadi et al., 2016) or the fluid flow in jointed rock or fault zones (Chen et al., 2000; Watanabe et al., 2008; Bisdom et al., 2016; Briggs et al., 2017; Jin et al., 2017; Zambrano et al., 2019; Kottwitz et al., 2019). However, the processes and parameters controlling the details of the fracture geometry are not fully understood yet.

Roughness can be defined as the deviation of a surface from a plane. The degree of roughness of a surface can be described in a number of different ways, ranging from visual, semi-quantitative approaches such as the "Joint Roughness Coefficient" (JRC) (Barton, 1973; Barton and Choubey, 1977) to fully quantitative measures derived directly from the geometrical properties of the surface such as the root mean square of the first deviation (slope) along a profile $Z_2$ (Myers, 1962) or the "structure function" SF proposed by Sayles & Thomas Sayles and Thomas (1977). It has been shown that the those measures are closely, but not perfectly, correlated to each other (Tse and Cruden, 1979; Li and Zhang, 2015). A roughness measure of particular interest due to its possible use in the parametrization of the fluid flow properties of rock fractures is the "effective surface area S" proposed

by Kottwitz et al. (2019), which can be considered as an extension of the "areal roughness index" defined by El-Soudani (1978) and therefore a 2D-analog of the "roughness profile index" defined there ($R_p$ in Li and Zhang (2015)) . A statistical analysis of rough surfaces shows that they can often be described as self-affine (Turcotte, 1992; Schmittbuhl et al., 1993, 1995; Bouchaud, 1997; Candela et al., 2009, 2012), i.e. they are statistically invariant under an affine transformation, but not under a global dilation (Bouchaud, 1997). In that case, the roughness can be described by a scaling parameter such as a fractal dimension or a Hurst exponent (Candela et al., 2009) in addition to a geometric roughness measure such as the root mean square deviation from an average plane at a given scale. While most of the previously mentioned parameters, i.e. JRC, $Z_2$, $R_p$ and SF, are measured along profiles across the surface, and are therefore intrinsically direction-dependent, the scaling parameters can be calculated either directionally or direction-independent.

Stress boundary conditions are one of the main factors controlling the shape and structure of faults and fractures in brittle rocks (Faulkner et al., 2010). While some experimental studies have investigated the dependence of the roughness of individual fracture surfaces on the stress conditions under which they were generated (Amitrano and Schmittbuhl, 2002), the use of numerical models makes it much easier to systematically study this issue for a wide range of stress parameters, including those which are difficult to access experimentally.

A large number of numerical modelling approaches has been developed to study the evolution and resulting properties of rough cracks, from statistical approaches like fiber bundle models over lattice methods including random fuse networks (RFN) and random spring networks (RSN) to standard continuum based approaches like finite element models (FEM) (Alava et al., 2006). In this work we use numerical simulations based on the Discrete Element Method (DEM) (Cundall and Strack, 1979; Donze et al., 1994; Mora and Place, 1994) to systematically study the formation of fracture surfaces under a wide range of stress conditions and to quantify their geometric properties. The focus of the investigation is on the initial geometry of the freshly formed fracture surfaces, i.e. in case of shear fractures, before significant slip takes place. This means that the results will be mainly applicable to joints and shear fractures with small displacement, both of which are very common structures in brittle rocks. The DEM approach was chosen due to its particular suitability for the numerical simulation of brittle deformation processes (Mair and Abe, 2008; Schöpfer et al., 2009; Schöpfer et al., 2011; Yoon et al., 2012) and the option to run true triaxial deformation experiments where $\sigma_1 > \sigma_2 > \sigma_3$, which are difficult to perform in the laboratory. In addition, we compare the results from the DEM simulations with data obtained from the photogrammetric analysis of fracture surfaces generated in triaxial compression experiments in the laboratory.

## 2 Method

### 2.1 Discrete Element Method

To simulate the process of rock fracture under an externally applied loading we are using the Discrete Element Method (Cundall and Strack, 1979; Donze et al., 1994; Mora and Place, 1994). In this approach a brittle-elastic material is modeled as a collection of spherical particles interacting with their nearest neighbors either by frictional-elastic interactions or by breakable elastic "bonded" interactions. Based on the force-displacement laws implemented in these interactions the forces and moments acting

on each particle can be calculated. The resulting translational and rotational accelerations of the particles are then used to calculate particle movements from Newtons equations. For the breakable bonded interactions a failure criterion is evaluated and if the failure threshold has been exceeded, the affected bonded interactions are removed and, if the particles involved are still in contact, replaced by a frictional-elastic interactions.

A range of different implementations exist for each of the interaction types, differing mainly in the details of the force-displacement law and, in case of the bonded interactions, the failure criterion. In this work we are using a linear force-displacement law for the normal component of the frictional-elastic interactions and a Coulomb friction law for the tangential component as described by Cundall and Strack (1979). For the bonded interaction we are using the bond model by Wang et al. (2006) which takes normal, shear, bending and torsional deformation into account. The stiffness and strength of the bonds are parameterized using the approach of Weatherley (2011) which calculates normal, shear, bending and torsional stiffness from the elastic parameters of an assumed bond material, specifically from Young's modulus $E_b$ and Poissons ratio $\nu_b$, considering cylindrical bonds with a length and diameter controlled by the radii of the particles they are connecting. A Mohr-Coulomb failure criterion is used for the bonds based on the strength parameters of the bond material, i.e. cohesion $C_b$ and friction angle $\Phi_b$.

Because the size of the individual models is important in this work to obtain high resolution roughness data from the simulated fracture surfaces, we are using the parallel DEM software ESyS-Particle (Abe et al., 2004), which enables the simulation of sufficiently large models.

## 2.2 Surface Extraction

The extraction of surface data from the numerical models requires two steps, (1) the identification of the individual fragments of the sample after fracturing (Fig 1b) and (2) the calculation which groups of the particles contained in each fragment form an individual fracture surface. The fragments of the broken sample are extracted by constructing an undirected graph from the structure of the DEM model such that the particles form the nodes of the graph and the remaining unbroken bonds form the edges in the graph. The fragments can then be extracted by calculating the connected components of that graph (Abe and Mair, 2005). For each fragment larger than $\approx 10\%$ of the original model (Fig. 1c) a ray-casting method is used to determine which of the particles are forming the surface of the fragment. In this approach, a set of parallel lines or "rays" with their origin outside the fragment and a specific direction is defined. The first intersection between each line and one of the particles is calculated using the algorithm proposed by Amanatides and Woo (1987). The positions of the calculated intersection points then form the point cloud describing the fragment surface (Figure 2).

To get a complete coverage of the fragment surface (Fig. 1e), i.e. to avoid shadowing effects by "overhanging" parts of the fragment surface, the calculations are performed for multiple view directions of the rays. The directions from the mass centers of all neighboring fragments to the mass center of the fragment and directions deviating from those by 30 degrees are used. To identify individual fracture surfaces two additional post-processing steps are performed. The initial outside surfaces of the intact model are removed by identifying each particle which was part of the surface of the model in the initial particle packing and removing the respective intersection points from the point cloud. In the final step a calculation is performed for

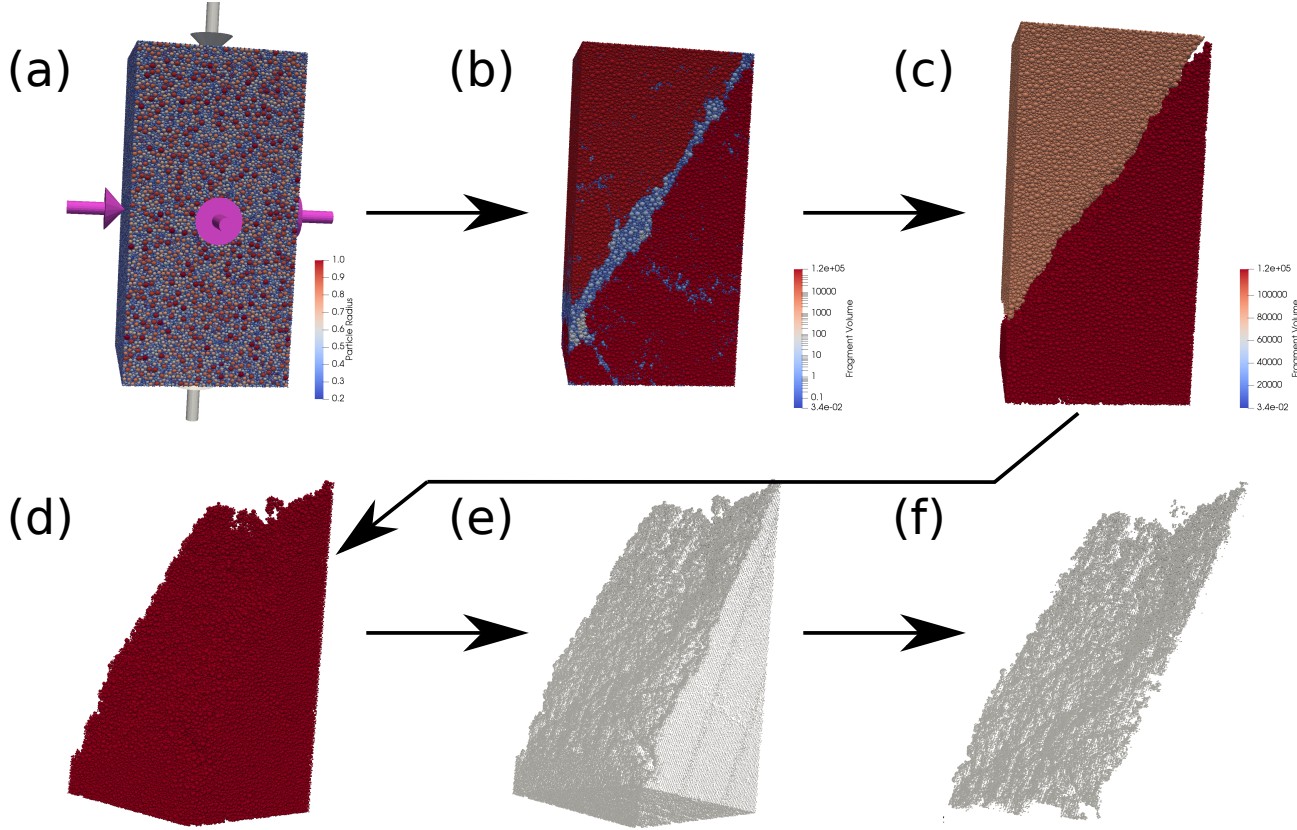

**Figure 1.** Numerical modeling workflow. (a) DEM specimen used for deformation experiments. Colors show particle size, purple arrows symbolize confining stress, gray arrows show compression direction. (b) Fragment identification in fractured DEM model. Colors show fragment size (red - large, blue - small). Red parts (top left & bottom right) show two major fragments, blue / white (i.e. fine grained) material along diagonal shows shear zone. (c) Two major fragments extracted from fractured DEM model. Colors show fragment size (volume). (d) Fragment extracted from DEM model. Rough fracture surface visible. (e) Point cloud surface generated from DEM model. Outer surfaces of the initial DEM specimen visible right and bottom. (f) Filtered point cloud used for analysis. Non-fracture surfaces and outlying points removed.

each particle contributing an intersection point to the surface point cloud to determine which other fragment is closest to this particle. This information is then used to split the point cloud into individual chunks, each representing an individual fracture

surface (Fig. 1f). By performing this step for all fragments in the model, corresponding pairs of surfaces belonging to the same fracture can be identified.

The 3D point clouds generated using this method are collections of $(x, y, z)$ coordinates. However, for most further analysis steps a representation of the surface as height field relative to a plane, i.e. as $z'(x', y')$ is needed. To obtain such a representation a "best fit" plane for the point cloud is calculated. The location of such a plane is found by calculating the barycenter $\boldsymbol{b} =$

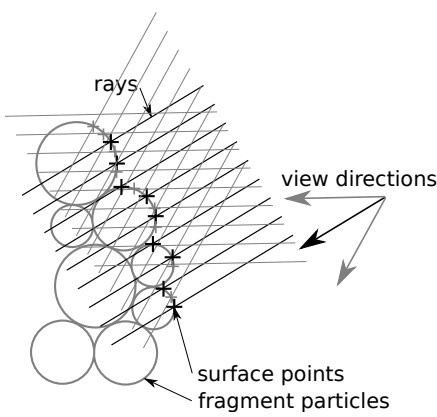

**Figure 2.** Simplified 2D sketch of the ray-casting method. The gray particles are assumed to belong to the same fragment of the deformed sample and the black and gray crosses show the fragment surface calculated from the line-particle intersections using multiple view directions. Black lines and black arrow show primary view direcion, light gray lines and arrows show additional view directions at a 30 degree angle to the primary direction.

$(x_0, y_0, z_0)$ of the point cloud, i.e.

$$(x_0, y_0, z_0) = \frac{1}{n} \sum_{i=0}^{n} (x_i, y_i, z_i) \tag{1}$$

and its orientation is determined by the two major eigenvectors $\boldsymbol{e}_1$ and $\boldsymbol{e}_2$ of the covariance matrix $\mathfrak{C}$ of the point cloud. The third eigenvector of the covariance matrix then determines the normal $\boldsymbol{n}_{fp} = \boldsymbol{e}_3$ of the plane. Using this, the in-plane coordinates $(x', y')$ of each point $\boldsymbol{p} = (x, y, z)$ and its perpendicular distance $z'$ from the plane can be calculated as

$$x' = (\boldsymbol{p} - \boldsymbol{b}) \cdot \boldsymbol{e}_1 \tag{2}$$

$$y' = (\boldsymbol{p} - \boldsymbol{b}) \cdot \boldsymbol{e}_2 \tag{3}$$

$$z' = (\boldsymbol{p} - \boldsymbol{b}) \cdot \boldsymbol{e}_3 \tag{4}$$

It should be noted that a surface can only be represented correctly as height field in this way if there are no parts of the surface which are "overhanging" with respect to the normal of the fitted plane, i.e. if there are no points on the surface with identical $(x', y')$ but different $z'$. However, this is generally the case for the fracture surfaces generated in the numerical models.

## 2.3 Roughness Characterization

A roughness measure commonly used in the study of the mechanical behavior of rock surfaces is the Joint Roughness Coefficient (JRC) defined initially as a parameter relating the shape of a rock joint to its peak shear strength, c.f. Eq. 9 in (Barton, 1973) or Eq. 2 in (Barton and Choubey, 1977). Its relation to the geometry of the joint surfaces was only qualitatively defined by assigning JRC values to a set of standard profiles (Barton and Choubey, 1977, Fig. 8). To estimate the JRC of an arbitrary

profile from measured geometrical data, a wide range of empirical formulas has been developed in the literature (Li and Zhang, 2015, Table 2). To calculate the approximate JRC of the fracture surfaces generated in the numerical models and the laboratory experiments three of the 47 equations presented there have been chosen. The subscript of the JRC in the equations below shows the respective number of the equation in (Li and Zhang, 2015, Table 2).

$$JRC_1 = 32.2 + 32.47 \log(Z_2) \tag{5}$$

$$JRC_{31} = 558.68\sqrt{R_p} - 557.13 \tag{6}$$

$$JRC_{34} = 92.97\sqrt{\delta} - 5.25 \tag{7}$$

where $R_p$ is the "Roughness profile index", $\delta = R_p - 1$ the "Profile elongation index" and $Z_2$ the "Root mean square of the first deviation of the profile", all as defined in (Li and Zhang, 2015, Table 1). $R_p$ is therefore calculated as

$$R_p = \frac{\sum_{i=1}^{N-1} \sqrt{(x_{i+1} - x_i)^2 + (y_{i+1} - y_i)^2}}{\sum_{i=1}^{N-1} \sqrt{x_{i+1} - x_i}} \tag{8}$$

and

$$Z_2 = \sqrt{\frac{1}{N} \sum_{i=1}^{N-1} \frac{(y_{i+1} - y_i)^2}{x_{i+1} - x_i}} \tag{9}$$

where $x_i$ is the abscissa of profile point $i$, $y_i$ its height above a mean value and $N$ is the number of sample points. Given that those parameters are calculated along profiles, the irregular point clouds generated using the method described in section 2.2 first need to be mapped to a regular grid.

Self-affine rough surfaces are characterized by the fact that they are statistically invariant under an affine transformation

$$(x, y, z) \rightarrow (ax, ay, a^H z) \tag{10}$$

where $x, y$ are the "in plane" coordinates of the surface and $z$ is the "height" of the surface above a given mean plane (Fig. 3a). The exponent $H$ is the Hurst exponent or "roughness index" (Mandelbrot and Van Ness, 1968; Mandelbrot, 1985; Bouchaud, 1997). A range of different method for the calculation of the Hurst exponent have been described in the literature (Renard et al., 2006; Candela et al., 2009), most of them either based on the evaluation of the power spectrum of the surface or correlation functions between the heights of points on the surface depending on their mutual distance. Because the point clouds describing the surfaces generated by the approach described in section 2.2 do not form a regular grid, spectral methods would require an additional interpolation step. Aside from the additional computational effort required, this might also introduce some difficult to quantify errors in the calculation of the Hurst exponent (Kottwitz, 2017).

In this work we therefore use the "Height-height Correlation Function" method as described by Candela et al. (2009). However, in contrast to the description in (Candela et al., 2009) the function is not calculated from 1D-profile data but directly from the 2D surface. The radially averaged Height-height correlation function is calculated as the root mean square (RMS)

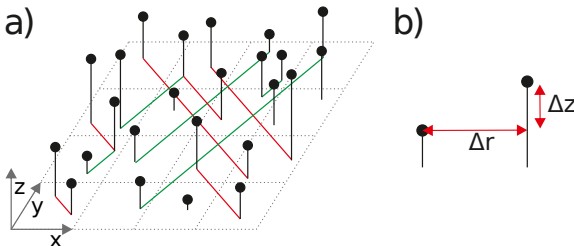

**Figure 3.** Height and distance relations of points in the point cloud used to calculate the height-height correlation function. (a) Arrangement of points above a fitted mean plane: dashed grid showing mean plane, black lines symbolizing orthogonal distance between plane and points and red / green lines showing relative orientation between points. (b) distance ($\Delta r$) and height difference ($\Delta z$) between points.

averaged height difference of all point pairs within a given distance range, i.e.

$$c(\Delta r) = \sqrt{\frac{1}{n} \sum_{\Delta r - w/2}^{\Delta r + w/2} \Delta z^2} \tag{11}$$

where $\Delta r = \sqrt{\Delta x^2 + \Delta y^2}$ is the "in-plane" distance between the points in the pair, $\Delta z$ is the height difference between the points (Fig. 3b), $w$ is the size of the distance bins over which the height differences are averaged and $n$ is the number of particle pairs in the respective distance bin. For the calculation of the angular dependence of the Height-height correlation function the direction between the two points of the pair is calculated as $\phi = \arctan(\frac{\Delta y}{\Delta x})$ and the summation of the height differences is adjusted from 1D distance bins in Eq. (11) to 2D (distance, direction) bins.

$$c(\Delta r, \phi) = \sqrt{\frac{1}{n} \sum_{\substack{\Delta r - w_r/2 \\ \phi - w_\phi/2}}^{\substack{\Delta r + w_r/2 \\ \phi + w_\phi/2}} \Delta z^2} \tag{12}$$

where $w_r$ is the bin size with respect to the in-plane distance of the points and $w_\phi$ is the bin size with respect to the direction from one point of the pair to the other. Due to the large number of points contained in the surface point clouds, which is as large $n_p \approx 500,000$ in some cases, and the resulting computational cost if all of the $n_p^2/2$ particle pairs would be taken into account, only a random sample of 10,000 points (i.e. $\approx 5 * 10^7$ point pairs) is used for each surface. Tests have shown that the reduction in the number of particle pairs evaluated does not impact the results.

Given that for a self-affine surface, the Height-height correlation function follows a power law, i.e. $c(\Delta r) \propto \Delta r^H$ (Candela et al., 2009) the Hurst exponent $H$ can be calculated by fitting a linear function to the straight part of the log-log plot of $c(\Delta r)$ vs. $\Delta r$. The slope of the linear function is the Hurst exponent. The Hurst exponent $H$ and the fractal dimension $D$ of an object are related as $D = 2 - H$ for a 1D-profile (Mandelbrot, 1985) or, more generally, $D = n + 1 - H$ where n is the dimension of the object (Yang and Lo, 1997), i.e. n=1 for a profile and n=2 for a surface.

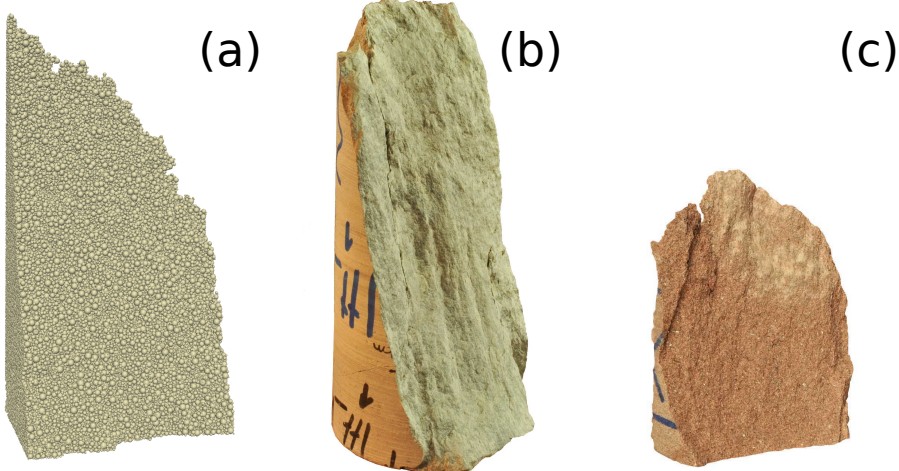

**Figure 4.** Types of fracture surfaces studied in this work. (a) numerical (DEM) model, (b) limestone fragment generated in triaxial deformation experiment, (c) sandstone fragment generated in uniaxial deformation experiment.

## 3  Experiments

A set of numerical simulations was performed to generate fracture surfaces under a wide range of stress conditions (Fig. 4a) and some natural rock samples were fractured in laboratory experiments (Fig. 4b,c). Both numerically and experimentally generated surfaces have been analyzed using the methods described in section 2.3.

### 3.1  Numerical Models

To generate a set of model fracture surfaces a large number of deformation experiments have been simulated. The set of simulations consists of unconfined compression ($\sigma_1 > 0$, $\sigma_2 = \sigma_3 = 0$), unconfined tension ($\sigma_3 < 0$, $\sigma_1 = \sigma_2 = 0$), standard triaxial compression ($\sigma_1 > 0$, $\sigma_2 = \sigma_3 > 0$) and true triaxial compression ($\sigma_1 > \sigma_2 > \sigma_3 > 0$) experiments. In all compressive models $\sigma_1$ is parallel to the y-axis, $\sigma_2$ is parallel to the x-axis and $\sigma_3$ is parallel to the z-axis, whereas in the unconfined tensile models $\sigma_3$, i.e. the extension direction, is parallel to the y-axis, $\sigma_1$ is parallel to the x-axis and $\sigma_2$ is parallel to the z-axis.

All models are using box-shaped samples with an aspect ration of $1 : 2 : 1$ contained between two servo-controlled plates in case of the unconfined compression and tension experiments or six servo-controlled plates for the standard triaxial and true triaxal experiments. While deformation experiments in the laboratory usually use cylindrical samples, we decided in favor of box-shaped samples because they make it much easier to apply the two different confining stresses in the true triaxial tests. In the tension experiments the plates are connected to the boundary particles of the sample by unbreakable bonds which only induce a force parallel to the normal of the plate but not perpendicular to it. This means the particle are free to move parallel to the loading plate, avoiding heterogeneous deformation ("necking"). In the compressive experiments both the axial loading

plates and, in the confined models, the plates along the x- and z-surfaces of the sample, interact with the boundary particles by
frictionless elastic interactions.

In the unconfined experiments ($\sigma_2 = \sigma_3 = 0$) a simple loading procedure is used, applying a prescribed displacement rate to the plates at the y-ends of the model to produce axial shortening or extension. During an initial phase the plate speed is ramped up smoothly according to a cosine function until the chosen speed is reached and then it is held constant for the main phase of the experiment. In the confined experiments($\sigma_2 \geq \sigma_3 > 0$) this loading procedure is preceded by a ramp-up of the stresses applied to the plates at the x- and z-sides of the sample until $\sigma_{xx} = \sigma_2$ and $\sigma_{zz} = \sigma_3$. For the smooth ramp-up of the applied stresses the same cosine function is used as for the ramp-up of the axial deformation rate in order to minimize unnecessary vibrations in the model. During this phase a stress is also applied to the loading plates at the y-ends of the sample such that $\sigma_1 = \sigma_2$. After a subsequent "rest" phase where the stress on all plates is held constant for given time to allow the particle movement introduced by the initial loading to dissipate, the same axial shortening as in the unconfined compression experiments is applied. A range of confining stresses from $\sigma_2 = \sigma_3 = 0$ to $\sigma_2 = \sigma_3 = 15$MPa was used for the numerical models in this work. In order to avoid the effect of abrasion modifying the roughness of the fracture surfaces after their initial formation, the state of the model immediately after one or more through-going fractures have formed was used for the extraction of fracture surfaces described in section 2.2. I.e. there is no, or at least very little, post-failure slip on those surfaces.

A model size of $55 \times 110 \times 55$ model units was chosen with a particle sizes ranging from $R_{min} = 0.2$ to $R_{max} = 1.0$, resulting in $\approx 950,000$ particles for those models (Fig. 1a). This model size was found in initial tests to provide a good balance between model resolution and computational cost. For the construction of the initial particle arrangement for the models the insertion based packing algorithm by Place and Mora (2001) was used. This algorithm generates dense particle packings having a power-law particle size distribution with an exponent of approximately $-3$, i.e. the number of particles with given radius $r$ is roughly proportional to $r^{-3}$.

In all deformation experiments the final loading plate speed was set to $\approx 17cm/s$. This is significantly higher than in real experiments, but using real lab values ($\mu m/s \ldots mm/s$) would lead to impractically long computing times because the time step of the calculations is restricted to values of $\Delta t \leqslant 3 \times 10^{-8}$s due to numerical stability constraints. Tests have shown that the increased velocities do not significantly influence the model results.

The mechanical properties of the DEM material have been calibrated to values similar to those of a typical sedimentary rock. The target values, Youngs Modulus E=30GPa and unconfined compressive strength UCS=80MPa, are within the range of sandstone or medium to high porosity limestone (Zoback, 2007). The failure strength was found to vary by less than 1% among samples. These parameters do not provide a direct match to the mechanical properties of the rocks used in the laboratory tests (Section 3.2), but the important ratio between failure strength of the material and the confining stress applied in the laboratory experiments lies well within the range covered by the numerical models (Fig. 5b). Because the details of the fracture behaviors of individual samples in DEM models show a well known dependence on the initial random particle arrangement (Koyama and Jing, 2007; Abe et al., 2011; Fakhimi and Gharahbagh, 2011), at least 5 simulations with different realization of the particle packing have been performed for each parameter set in order to quantify this variability.

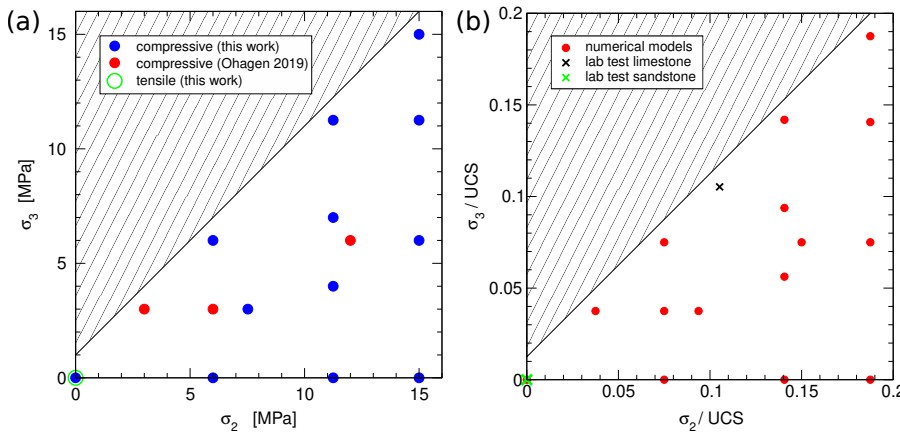

**Figure 5.** Confining stress range covered by the numerical models, combining the experiments in this work and the data from (Ohagen, 2019). (a) Numerical models only, using absolute stress values. (b) Numerical and laboratory experiments, using stress values scaled by the unconfined compressive strength of the respective material. Hatched segment in top left of the diagrams: parameter space excluded by the condition $\sigma_3 \leq \sigma_2$.

To improve the coverage of the chosen range of stress conditions, data from a related study (Ohagen, 2019) was integrated into the analysis (Fig 5a). This study was using an identical model setup, except for slightly smaller models with dimensions of $40 \times 80 \times 40$ model units ($\approx 360,000$ particles) and $50 \times 100 \times 50$ model units ($\approx 710.000$ particles) compared to the roughly 950,000 particles used in most models in this work.

## 3.2 Laboratory Tests

To compare the roughness of fractures created in the DEM models with the roughness of real fractures we conducted a number of laboratory uniaxial and triaxial deformation experiments. For our study we used a suite of fine grained, low porosity Upper Jurassic carbonate rock samples and additionally one Lower Triassic sandstone sample, both from Franconia, Germany. Sample size for the experiments were 55x110mm cylinders. The main goal of the experiments was to produce fractures for given stress conditions which could then be used for roughness analyses.

The sandstone uniaxial compressive strength (UCS) experiment lead to an typical hourglass fracture pattern, splitting the sample into a small number of larger fragments, which could be used for further analyses (Fig. 4c). Unfortunately, for the UCS and most of the triaxial experiments of the carbonate rocks the samples disintegrated into a very large number of very small fragments leaving no suitable fracture surfaces to analyze. See Fig. S1 in supplement for a typical example. This applied in particular to the samples loaded with small confining pressures. Only in one experiment with a confining pressure of 30 MPa post-deformation fragments were large enough for our planned fracture surface analyses (Fig. 4b). From the suitable fragments we constructed a digital three dimensional surface model using photogrammetric methods. The models were built from more than 100 single pictures of the samples from different perspectives using a 12 megapixel SLR camera and a 40mm macro lens.

The photos were taken from a distance of 5cm to 10cm between front lens and the object, which is close to the minimum focus distance of the lens used. The models were then clipped to the fracture plane of interest. The remaining surface geometry was exported as 3D point cloud data with c. 2.2 million data points in total, resulting in a point density of c. 28000 points/cm$^2$ and an average point distance of 60 micrometers.

The generated point clouds were then used for roughness analyses of the fracture surfaces following the approach described in section 2.3. Besides the creation of fracture surfaces the deformation experiments were also used to derive typical geomechanical properties of the carbonate and sandstone samples which were used for comparison with the DEM models. For the carbonate rocks a UCS of $\approx 285$MPa was obtained and $\approx 85$MPa for the sandstone sample. For the limestone a friction coefficient $\mu = 0.7$ was derived from experiments with confining pressures ranging between 0 to 30 MPa. Young's modulus was measured at $E = 48$GPa for the limestone and $E = 12.5$GPa for the sandstone.

## 4 Results

### 4.1 Numerical Models

Based on the data produced by a total of 131 numerical simulations the geometrical properties of 388 fracture surfaces have been analyzed. The fracture orientations were as expected under the stress conditions. The dip angle was typically within 25-35 degrees of $\sigma_1$, i.e. 55-65 degrees assuming $\sigma_1$ to be vertical. The strike direction of the majority of the fractures was within $\approx$ 10 degrees of $\sigma_2$ in the true triaxial models ($\sigma_2 \neq \sigma_3$) and more or less randomly distributed under transverse isotropic stress conditions ($\sigma_2 = \sigma_3$).

In an initial step the joint roughness coefficients for a small set of surfaces were approximated using Eq. (5) - (7). The results did show that the resulting JRC values were consistently above the range defined by Barton and Choubey (1977), i.e. larger than 20, and therefore also outside the range of validity of the approximation equations in Li and Zhang (2015). Similarly, the geometric parameters $R_p$ (Eq. 8) and $Z_2$ (Eq. 9) from which the estimated JRC values were calculated, were outside the applicable ranges given there. While the roughness produced by the numerical models is therefore outside the range for which the fitting equations collected by Li and Zhang (2015) were originally intended, Figure 1a in their work suggests that Eq. (5) would be the best option to extend the range of approximate JRCs to the surface geometries observed here because it provides a particularly good fit at large values (i.e. $Z_2 \approx 0.35 - 0.4$, $JRC \approx 20$). Therefore Eq. (5) was used to estimate the average JRC for each of 261 surfaces based on a total of $\approx 24400$ profiles. The remaining 127 of the 388 surfaces studied were found to be too small in at least one of the dimensions to allow the extraction of sufficiently long profiles. For each surface, profiles were generated in two orthogonal directions to check for a possible anisotropy of the surface roughness. The results did show that the mean estimated JRCs for the profiles differs by less than 10% between the two direction, which is generally less than the standard deviation between the profiles within one direction. Plotting the estimated JRC for the analyzed surfaces against the mean confining stress of the models (Fig. 6a) shows that there is no clear trend of JRC vs. confinement, but that models with transversely isotropic confinement ($\sigma_2 = \sigma_3$) generally have higher JRC values than models fractured under true triaxial

conditions, i.e. $\sigma_2 \neq \sigma_3$. The directly calculated geometric roughness measures, i.e. $R_p$ and $Z_2$, show a very similar pattern (Fig. 6b,c).

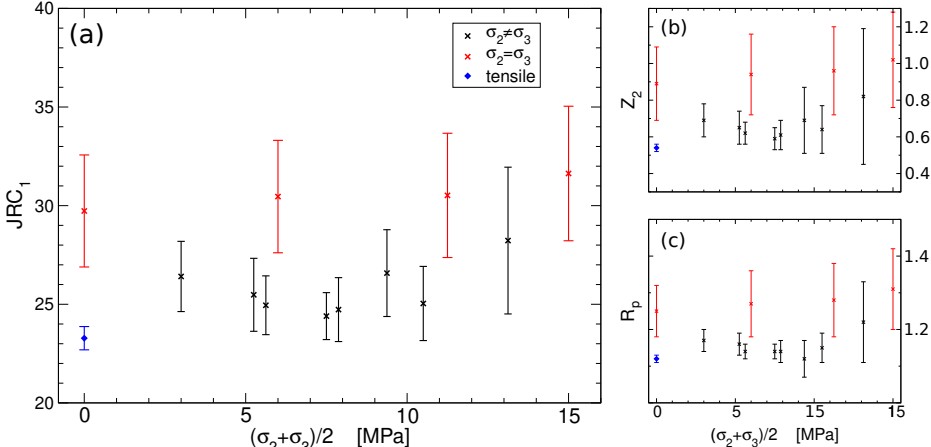

**Figure 6.** Geometric roughness measures for surfaces generated at different stress conditions in DEM models. Black: True triaxial compression, bed: Standard triaxial compression (transverse isotropic confinement), blue: unconfined extension. (a) Approximated JRC values calculated based on Eq. (5), (b) "Root mean square of the first deviation" $Z_2$ (Eq. 9), (c) "Profile elongation index" $R_p$ (Eq. 8). Error bars show standard deviation.

The perpendicular distance or "height" of the points of the fracture surfaces above a fitted fit plane is calculated according to Eq. (4). Analysis shows that the heights are normally distributed (Fig. 7) as expected for fracture surfaces (Ponson et al., 2007), allowing a "RMS roughness" $h_{rms} = \frac{1}{n}\sqrt{\sum_n(z'^2)}$ to be calculated.

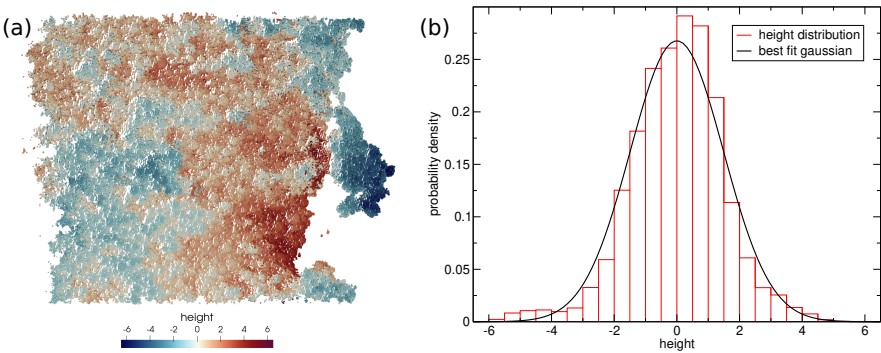

**Figure 7.** Distribution of heights of a simulated fracture surface above a "best fit" plane. Data taken from model with $\sigma_2$=5MPa and $\sigma_3$=0MPa. (a) Map view of the surface colored by height above the "best fit" plane. (b) Probability density of heights and fitted normal distribution.

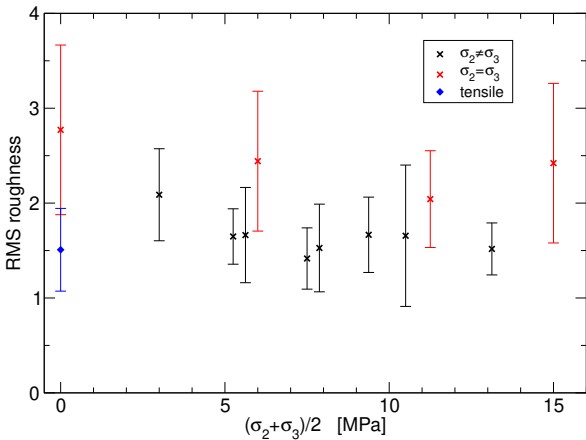

**Figure 8.** Average RMS roughness values for surfaces generated at different stress conditions. Black: True triaxial compression, Red: Standard triaxial compression (transverse isotropic confinement), Blue: unconfined extension. Error bars show standard deviation.

Plotting the RMS roughness $h_{rms}$ of all models against the mean confining stress (Fig. 8) shows that there is no clear
dependence between the two parameters, except for a difference between transverse isotropic ($\sigma_2 = \sigma_3$) and true triaxial ($\sigma_2 \neq \sigma_3$) stress conditions. In case of the transverse isotropic confinement the observed RMS roughness $h_{rms} = 2.35 \pm 0.78$ model units is higher than in case of true triaxial conditions where $h_{rms} = 1.63 \pm 0.48$ model units. It can also be observed that the RMS roughness of the models subjected to unconfined extension (blue marker in Fig. 8) is smaller at $h_{rms} = 1.51 \pm 0.44$ model units than that of the models subjected to unconfined compression with $h_{rms} = 2.76 \pm 0.88$ model units. This difference, however, is possibly at least in part an artifact of the different size of the fracture surfaces between the two model groups. In the tensile case, the fractures tend to be roughly normal to the extension direction, i.e. the long axis of the model and their size is therefore restricted to the small cross section of the model. In contrast, the fracture surfaces in the compressive case tend to be oriented such that their normal is at an angle of $\approx 55 \ldots 60$ degrees to the compression direction and can therefore grow as large as a plane diagonally across the model, i.e. more than twice the size compared to the tensile case. Plotting the height-height correlation function (Eq. 11) of the surfaces in a log-log plot (Fig. 9) shows a clear linear section which, for most surfaces analyzed, ranges from $\Delta r_{min} \approx 1.5 - 2$ model units, i.e. somewhat more than the maximum particle size, to about half of the smaller dimension of the surface, which in most cases means $\Delta r_{max} \approx 20 - 30$ model units. This linear section in the log-log plot, representing a power-law dependence $c(\Delta r) \propto \Delta r^h$ shows that the surface is indeed self affine, at least for the range of scales covered by the linear section.

In order to verify that the observed self-affine structure of the fracture surfaces generated in the numerical model is indeed a result of the fracture process, and not an artefact caused by the intrinsic roughness of surfaces in the particle model, a number of "quasi-planar" surfaces was generated in the particle model and their roughness was analysed. For this purpose one of the blocks of packed particles used in the DEM simulations of the triaxial tests (Fig. 1a) was cut with an arbitrarily oriented plane, i.e. the particles on one side of the plane were removed. The remaining fragment of the block then underwent the same surface

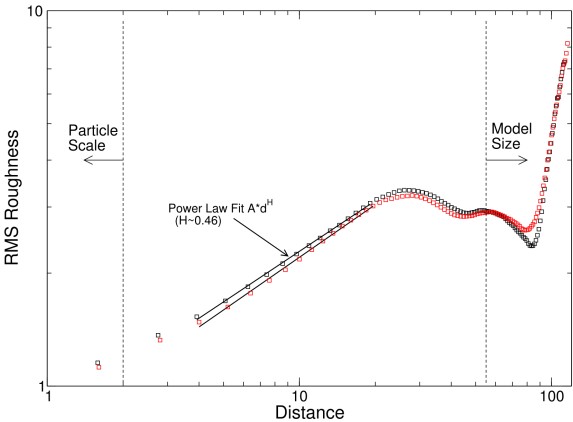

**Figure 9.** Log-log plot of the height-height correlation function of the two surfaces of a single fracture. Red and black symbols show the RMS height differences calculated for each distance bin for the two surfaces. The straight lines are fitted to the linear section of the data in log-log space, showing a power-law dependence.

extraction and roughness analysis procedures as the fracture surfaces produced in the deformation experiments. The result (Fig. 10) shows that the height-height correlation function of the cut surface is essentially flat from the particle scale up to the model size. Performing this analysis on multiple cut surfaces did show that this is independent of the orientation of the cut plane and the details of the particle packing. Only the absolute value of the roughness of the cut surfaces depends somewhat on the size range of the particles. Calculating the joint roughness coefficients for the cut surfaces according to Eq. (5) did, as expected, produce non-zero values of the JRC. However, the JRC-values for the cut surfaces are in the range of 11.5-12, which is much smaller than the values observed in the fracture surfaces generated in the numerical models (JRC $\approx 23-32$, Fig. 6a). It can therefore be assumed that, while there is some contribution of the intrinsic particle scale roughness to the total roughness of the fracture surface, the self-affine structure of the fracture surfaces as well as the major part of their total roughness is due to the fracture process and not the particle structure of the model as such.

Performing the calculations for all 388 fracture surfaces extracted from the numerical models produced Hurst exponents ranging from 0.2 to 0.6. To investigate possible dependencies on the stress conditions under which the fractures were created, the average Hurst exponents over all surfaces generated in each set of simulations with identical boundary conditions have been calculated. The mean value of $H$ for the sets varies between 0.3 and 0.45, with the variation between top and bottom quartile within each set typically in the range of $0.05\ldots0.1$. Due to the relatively small number of data points within each set of models, i.e. between 8 and 28 surfaces, and the observed asymmetry of the error distribution in some instances, quartiles have been calculated and plotted (Fig. 11 and Fig. 12) instead of standard deviations. Plotting the calculated Hurst exponents against the mean confining stress $(\sigma_2 + \sigma_3)/2$ (Fig. 11) shows a weak trend towards lower Hurst exponents with increasing confinement. No dependence of the Hurst exponent on the ratio between the confining stresses $\sigma_3/\sigma_2$ could be observed (Fig. 12).

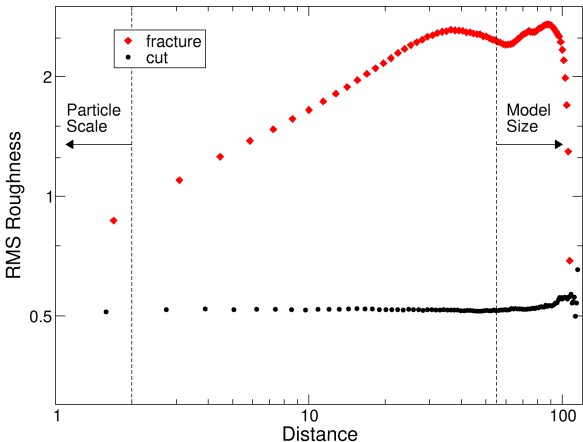

**Figure 10.** Log-log plot of the height-height correlation function of a fracture surface generated in a numerical deformation experiment (red diamonds) and a "quasi-planar" surface generated by cutting the particle packing used in the experiment with a plane (black circles).

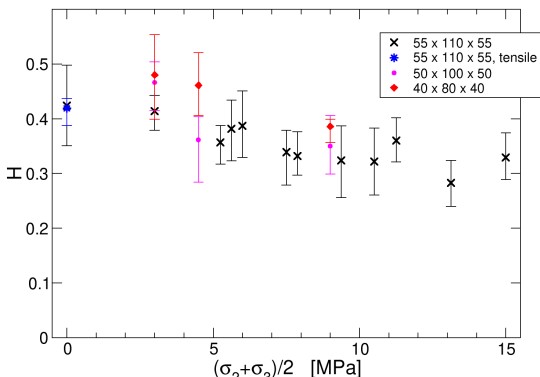

**Figure 11.** Average and variability of Hurst exponents for surfaces generated with different average confining stress $(\sigma_2 + \sigma_3)/2$. Black: triaxial compression models, this work, Blue: unconfined extension models, this work, Red and pink: triaxial compression models, data from Ohagen (2019). Error bars show top and bottom quartile.

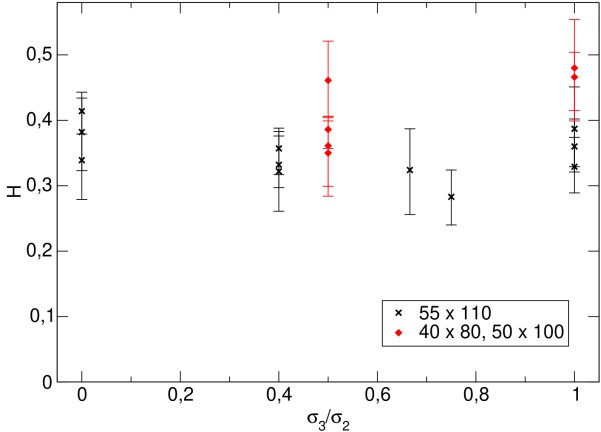

**Figure 12.** Average and variability of Hurst exponents for surfaces generated with different ratio of confining stresses $\sigma_2$ and $\sigma_3$. Black: this work, Red: data from Ohagen (2019). Error bars show top and bottom quartile.

## 4.2 Laboratory Tests

To characterize the roughness of the fracture surfaces produced in the laboratory deformation tests, we examined the photogrammetrically produced point clouds of the single sample fragments. For each of the limestone and sandstone samples one fracture surface was chosen. The maximum sampling area for the roughness investigation was $14cm^2$ for the sandstone and $\approx 25cm^2$ for the limestone sample. The analyses of the heights distances of the single points of the point clouds above their fitted mean planes revealed a normal distribution of the heights. Thus, a calculation of the RMS roughness is justified (Fig. 13). The height-height correlation functions of these surfaces have a well-defined linear section in a log-log plot proving a self-affine geometry in a distance range between $\approx$0.1cm and $\approx$1cm, both for the limestone and sandstone sample (Fig. S2 and S3 in the supplement). With distances larger $\approx$1cm a flattening of RMS curve can be observed, marking the upper end of the power-law relationship between the point distance and the RMS height difference. From the linear slope segments of the correlation functions similar Hurst exponents could be deduced with H=0.66 for the sandstone and H=0.69 for the limestone when analysing the maximum sampling area on the respective fracture.

To check whether the size of the investigation area on the fracture surfaces has an effect on calculated Hurst exponents we analyzed the height-height correlation functions and Hurst exponents for a suite of different area sizes (Fig. 14). For the limestone sample the mean H-value results in H=0.73 with a standard deviation of 0.08. The sandstone sample shows a clearly lower mean H-value of H=0.6 and a standard deviation of 0.05. A stronger scatter of Hurst exponents can be observed for the smallest analyzed sample area size of $\approx 1cm^2$, ranging between H=0.43 and 0.64 for the sandstone surface and between H=0.67 and 0.85 with two outliers of $H \approx 0.5$ for the limestone surfaces. For these outliers a closer investigation of the corresponding RMS/Distance curves shows that two different linear sections could be derived, one with a higher Hurst exponent for smaller distances and one lower Hurst exponent for larger distances.

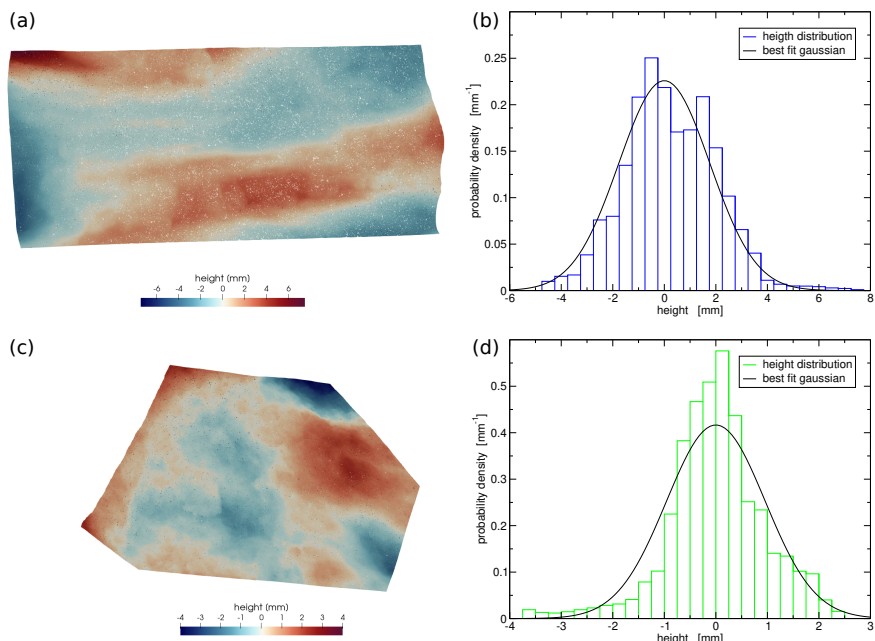

**Figure 13.** Distribution of heights for fracture surfaces generated in laboratory deformation experiments. Top: limestone sample deformed at $\sigma_2 = \sigma_3$=30MPa, bottom: sandstone sample deformed in unconfined compression test. (a) Map view of the limestone surface colored by height above the "best fit" plane. (b) Probability density of heights and fitted normal distribution for limestone surface (c) Map view of the sandstone surface colored by height above the "best fit" plane. (d) Probability density of heights and fitted normal distribution for sandstone surface.

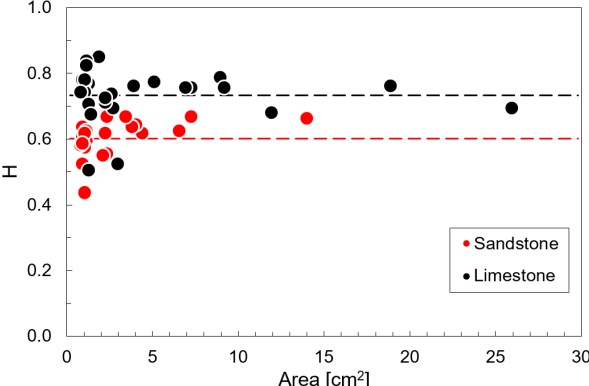

**Figure 14.** Calculated Hurst exponent for different sizes of the measurement area in the natural rock samples. Circles: individual measurements, dashed lines: average of all measurements per sample.

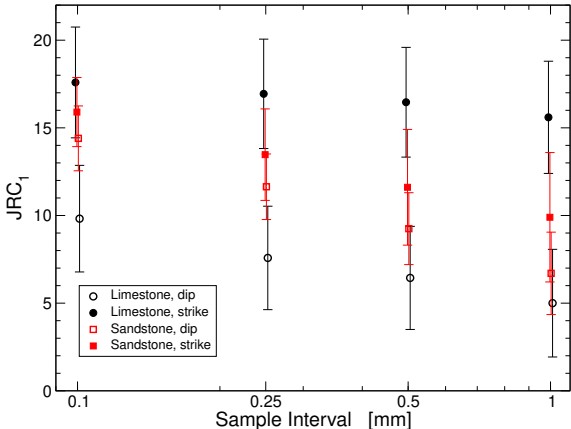

**Figure 15.** Estimated JRC values calculated based on Eq. (5) for fracture surfaces of the sandstone and limestone specimen. Black: Limestone, red: Sandstone. Open symbols: profiles taken parallel to shortening direction, filled symbols: profiles perpendicular to shortening direction. Small horizontal offset between data points in each group added for better visibility of individual error bars.

For both sample surfaces the joint roughness coefficient (JRC) was estimated using the same methods as for the numerical models. The results show that the estimated JRC is dependent on the sampling resolution, i.e. the number of sampling points on the profile, specifically that the calculated value of the JRC is increasing with smaller sampling intervals (Fig. 15). This is a known effect, which is caused by the dependence of the underlying geometric parameters $R_p$ and $Z_2$, from which the estimated JRC is calculated, on the sampling interval used (Yu and Vayssade, 1991). It is also to be expected based on the fact

that the analyzed surfaces are self-affine. In that case the dependence of $R_p$ on the sampling interval is directly described by the "compass dimension" (Mandelbrot, 1985) of the profile. For the fracture surfaces in the numerical models (Section 4.1) a similar analysis of the resolution dependence of the JRC was not done because of the lower intrinsic resolution of the point clouds which limits profiles to less than 100 sample points in most cases.

    The empirical equations used for the calculation of JRC from measured geometric parameters are usually derived based on

sampling resolutions between 100 and 400 points per profile (Tse and Cruden, 1979; Yu and Vayssade, 1991; Li and Zhang, 2015). Specifically, the equation used in this work to estimate JRC from $Z_2$ (Eq. 5) was derived by Tse & Cruden (Tse and Cruden, 1979) using a sample interval of 1.27mm at a profile length of 25cm, i.e. slightly less than 200 points. The surfaces analyzed here have dimensions of about 7cm × 5cm for the sandstone and approximately 10cm × 4.5cm for the limestone. Therefore a sampling interval of between 0.25mm and 0.5mm will produce a similar number of sample points along the profiles.

Therefore the best estimates for the average JRC of the fracture surfaces produced in the laboratory experiments are for the sandstone $JRC \approx 9-11$ in the direction parallel to shortening direction in the deformation experiment and $JRC \approx 11-13$ perpendicular to it (Fig. 15). For the limestone the estimates are $JRC \approx 6.5-7.5$ in the parallel direction and $JRC \approx 16-17$ in the perpendicular direction. In both cases the JRC shows a clear anisotropy between the two directions. However, this anisotropy is much larger in the limestone compared to the sandstone sample.

| particle size range | Hurst exponent | JRC | JRC anisotropy |
|---|---|---|---|
| 0.2-1.0 | 0.414±0.5 | 26.1±1.8 | 2.0% |
| 0.15-1.0 | 0.415±0.85 | 25.4±2.4 | 3.2% |
| 0.1-1.0 | 0.398±0.96 | 24.2±2.3 | 0.4% |

**Table 1.** Roughness properties for surfaces generated in numerical simulations of triaxial compression tests at $\sigma_2 = 6$MPa, $\sigma_3 = 0$ using different particle size ranges for the DEM material.

## 5 Discussion

The results of the analysis of the simulation data (Section 4.1) shows that the roughness of the fracture surfaces generated in the numerical models is high compared to natural rock fractures usually considered in the geomechanical literature. In the numerical models the surfaces show estimated JRC values larger than 23 and in some case exceeding 30 whereas the JRC for natural surfaces was originally only defined for a range up to 20 (Barton, 1973; Barton and Choubey, 1977). In contrast, the natural rock samples analyzed in this work (Section 4.2) show JRC values between 6 and 17 which is well within the range defined by Barton (1973).

However, as described in section 4.1 the JRC values for the numerical model contain a small contribution due to the intrinsic particle scale roughness of the model. If we consider that the total roughness of the surface is the sum of the roughness due to the particle structure of the surfaces and the roughness due to the actual fracture process, and if we assume that those contributions are not spatially correlated with each other, it would be possible to correct the calculated JRC-values by removing the effect of the particle scale roughness. The parameter $Z_2$ on which the calculation of the JRC is based (Eq. 5) is calculated from the RMS of the first derivative of profiles along the surface (Eq. 9). Based on the assumption that the particle-scale roughness and the fracture-generated roughness are not spatially correlated, this means that the total $Z_2$ is the RMS of the $Z_2$-values of the two parts, and therefore the value $Z_{2f}$ of the facture-generated roughness can be estimated as $Z_{2f} = \sqrt{Z_2^2 - Z_{2p}^2}$ where $Z_{2p}$ is the contribution of the particle-scale roughness. Using the data described in section 4.1, values of $Z_{2p} \approx 0.23 - 0.24$ are obtained. This would result in a correction of the mean JRC-values for the different groups of surfaces shown in Fig. 6a from $\approx 23.7$ to $\approx 22.1$ for the smallest and from $\approx 32.2$ to $\approx 31.8$ for the largest values of the JRC. This shows that the potential corrections are not significant and, in most cases, well inside the scatter of the calculated JRC values. In addition, we did run 2 small sets of simulations using a wider range of particle sizes than the "standard" models described in section 3.1, i.e. a larger ratio between maximum and minimum particle radius ($R_{max} : R_{min} = 1.0 : 0.15$ and $R_{max} : R_{min} = 1.0 : 0.1$), to see if the particle size range had any effect on the surface properties. These sets consisted of 5 simulations each, all performed under true triaxial conditions using $\sigma_2 = 6$MPa and $\sigma_3 = 0$. The results did not show a statistically significant difference in Hurst exponent or JRC compared to the equivalent simulations performed using the particle radius range $R_{max} : R_{min} = 1.0 : 0.2$ (Tab. 1).

In numerical models there is a slightly higher anisotropy in the models with transversely isotropic confinement ($\sigma_2 = \sigma_3$) of up to 8% difference in JRC between the directions whereas in the models with $\sigma_2 \neq \sigma_3$ the difference is less than 3%

in all cases. In the rock samples, which are also deformed under conditions where $\sigma_2 = \sigma_3$, the anisotropy is much higher, i.e. the ratio between the JRC in the two directions is $\approx 1 : 1.2$ in the sandstone and $\approx 1 : 2.3$ in the limestone. However, due to small number of fracture surfaces available for analysis from the laboratory experiments it is not clear if this strong anisotropy, and the large difference between the limestone and the sandstone sample, is a general property of fracture surfaces generated under comparable conditions or just an artifact of the specific samples studied. In general the strong anisotropy which was observed in the laboratory experiments, in particular in the limestone, was not replicated in the numerical models. The reason for the stronger directional anisotropy in the natural rock samples is not clear yet. A key difference between the micro-scale mechanics of laboratory and numerical experiments is that the natural rocks can undergo grain size reduction during the fracture process whereas this mechanism is not implemented in the numerical models used in this paper. This might explain why the numerical models, at least in our experiments, do not produce the striations observed in the natural rock samples. A possibility to test this hypothesis in future work would be to extend the numerical models to use breakable particle clusters to represent rock grains instead of single particles. This approach has been shown to yield insights into the micro-mechanics of grain size reduction processes, for example in fault gouge (Abe and Mair, 2005; Mair and Abe, 2008, 2011) and in compression experiments (Thornton et al., 2004). However, it also significantly increases the required computational effort for the simulations. A computationally less expensive option to include grain size reduction into the numerical models might be to adapt the empirical particle replacement approach developed by Cleary (2001) to the specific requirements of the simulation of rock fracture under triaxial loading. However, as Weerasekara et al. (2013) point out, this approach is strongly dependent on the availability of good calibration data for the grain fracture under the specific stress and strain rate conditions of the process modelled. Further insights could also be provided by additional laboratory experiments, for example to test if the difference in anisotropy between numerical and laboratory experiments also exists under true triaxial conditions ($\sigma_2 \neq \sigma_3$).

Smoothing due to abrasion while sliding is, in general, an important mechanism for the modification of rough surfaces. In particular, slip along the surface can result in a significant reduction of the Hurst exponent for profiles parallel to the slip direction down to values below 0.5 (Candela et al., 2012, Table 1b). However, those large reductions appear to apply mainly to faults with large amounts of slip, i.e. several meters up to kilometers. In contrast, data from laboratory experiments published in the literature (Amitrano and Schmittbuhl, 2002; Davidesko et al., 2014; Badt et al., 2016) suggest that this process is unlikely to have a sufficiently large effect at the small shear offsets in both numerical models and experimental samples studied here to explain the observed differences. To investigate if the roughness evolution of the fracture surfaces with increasing deformation of the sample plays a role in our numerical model we did perform a small number of simulations which did not stop immediately after the formation of the fractures, but instead continued deformation to a total axial strain of up to 12%. This is significantly larger than the strain occurring in the laboratory experiments, where total axial shortening did not exceed about 2%. In particular, the amount of shortening occurring after the peak axial stress was reached, i.e. after failure, was generally less than 1%. The obtained Hurst exponents did show no significant trend with increasing strain of the model and offset of the shear fracture (Fig. S4 in the supplement). While the average of the Hurst exponents from the 6 surfaces investigated could be considered as showing a slight increasing trend for axial strains up to 8% (Fig. S5 in the supplement), the increase of 0.03 is about an order of magnitude too small to explain the observed differences between numerical and experimental surfaces.

However, it would be compatible with the effect observed by Amitrano and Schmittbuhl (2002). For one of the models we did also calculate the JRC of the surfaces at various stages of the simulation. The data shows that there is also no significant change of the JRC for the shear offset considered in this model, which would be equivalent to $\approx 1$cm in the laboratory samples, and under the conditions of this model, i.e. true triaxial stress with $\sigma_2 = 7.5$MPa, $\sigma_3 = 3$MPa (Fig. S6 in the supplement). This seems to confirm again that under the small shear offsets relevant for our experiments, there is very little evolution of the surface roughness, at least as far as it concerns the roughness parameters calculated here (Hurst exponent, JRC). In particular, the data would suggest that any effects due to the small, but the non-zero, shear offset in the laboratory experiments are much too small to explain the observed differences between numerical simulations and laboratory experiments.

Based on the results from the numerical models there appears to be a trend towards higher roughness for fracture surfaces generated under transversely isotropic stress conditions, i.e. standard triaxial compression ($\sigma_1 > \sigma_2 = \sigma_3$) compared to those generated under true triaxial conditions ($\sigma_2 \neq \sigma_3$). This trend was shown for both geometrical roughness measures used in the analysis of the data from the numerical experiments, i.e. the joint roughness coefficient JRC (Fig. 6) and also the RMS roughness (Fig. 8). A possible, but at this stage purely speculative, idea to explain this observation might be that, if we assume that the through-going fractures, which we analyze, form by coalescence from smaller, precursory, fractures, those precursory fractures have their strike angles constrained to a narrow range if $\sigma_2 \neq \sigma_3$, but that there is no such constraint if $\sigma_2 = \sigma_3$. If this is the case, then the coalescence of those precursory fractures might lead to smoother large-scale surfaces if they all have similar orientations compared to when they have random strike directions. Unfortunately the numerical models used in this work do not have the resolution necessary to test this hypothesis.

Additionally, a difference in the roughness between the surfaces on tensile and compressive (i.e. shear-) fractures generated under unconfined conditions has been observed, with the tensile fractures showing a smaller roughness. This effect appears to be more pronounced if the roughness is measured in terms of the JRC compared to the RMS roughness. Should these effects be confirmed by further work, and in particular by comparison with more experimental data, it could be used to provide additional input data to, for example, permeability estimations of fracture networks or geomechanical fault stability calculations.

The analysis of the roughness scaling properties of the surfaces in terms of the height-height correlation function shows that the fracture surfaces generated in the numerical models are self affine with Hurst exponents around 0.3 - 0.45. This value is in disagreement with the majority of field and experimental studies (Bouchaud et al., 1990; Schmittbuhl et al., 1993, 1995; Bouchaud, 1997) which find a "universal" Hurst exponent $H \approx 0.8$. However, low Hurst exponents in the range $H \approx 0.4 - 0.5$ have previously also been found in other numerical models of the generation of rough fractures such as 3D random fuse networks (Alava et al., 2006).

The Hurst exponents of the surfaces generated in the numerical models can be corrected for the influence of the particle scale roughness in a similar way to the procedure described above for the correction of the joint roughness coefficients. It would require correcting the RMS roughness values in the height-height correlation function for each individual distance bin and obtaining a power-law fit based on the corrected data points (Fig. 16). However, while these corrections do lead to slightly higher calculated Hurst exponents, the increase is at most about 0.05 and therefore the effect is far too small to explain the discrepancy.

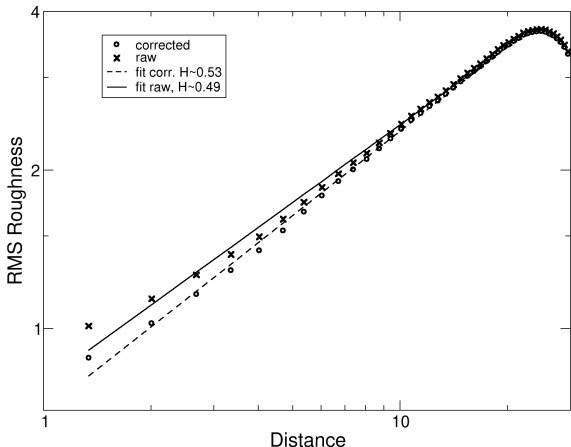

**Figure 16.** Comparison of the height-height correlation functions of a numerical fracture surface based on raw data (crosses) and corrected for the influence of the particle-scale roughness (circles). Lines are power law fits used to calculate the Hurst exponents for raw (continuous line, $H \approx 0.49$) and corrected data (dashed line $H \approx 0.53$).

The data obtained from the fracture surfaces generated in triaxial tests on the limestone sample ($H \approx 0.75$) are compatible with this "universal exponent". In contrast, the sandstone sample shows a lower Hurst exponent ($H \approx 0.6$) than the limestone sample, but not as low as the numerical models. There is experimental data for sandstone in the literature showing Hurst exponents even lower than our sandstone sample and in fact close to the results from the numerical models, i.e. $H = 0.47\pm0.04$, (Boffa et al., 1998) and similar data from a synthetic, sandstone-like material made from sintered glass beads ($H = 0.40\pm0.04$, Ponson et al. (2006)). Both those studies investigated tensile (mode-1) fractures. Boffa et al. (1998) used a direct tension setup with a pre-notched sample to initiate the fracture at a defined location whereas Ponson et al. (2006) used a modified Brazilian test where a compressive load is applied to two opposite points on the circumference of the cylindrical sample to generate a tensile stress in the stress in the central part of the disk (Jaeger et al., 2007; Fjaer et al., 2008). However, our numerical models do not show a dependence of the Hurst exponent on the fracture mode (Fig. 11).

Nigon et al. did observe a transition from a Hurst exponent of 0.74 to a lower value of 0.5 below a length scale of about 0.1mm in natural joint surfaces in sandstone (Nigon et al., 2017, Figure 9). However, this transition scale from a "jointing induced roughness" to a "grain induced roughness" is at a scale comparable to the mean grain size in their material. The equivalent length scale in our numerical models would be the mean particle diameter, i.e. below 1 model unit, which is well below the length range used to fit the scaling law (Fig. 9). This difference in scales shows that the Hurst-exponents in our numerical models are completely calculated above the "transition scale" of Nigon et al. (2017) and therefore should belong to the regime described as "jointing induced roughness" by them. This means that the low values of the Hurst-exponents in the numerical can not be explained by the "grain induced roughness" regime of Nigon et al. (2017).

When comparing the data from the numerical models to the relation between fractal dimension $D$ and JRC proposed by Ficker (2017), i.e. $\mathrm{JRC} \approx 50(D-1)$, the surfaces show on average a slightly smaller JRC than would be expected based on

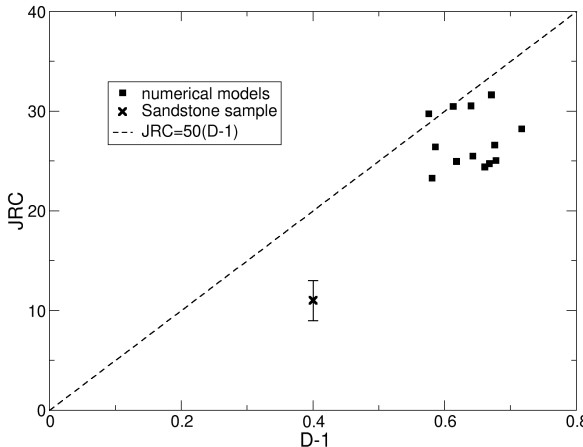

**Figure 17.** Relation between joint roughness coefficient and Hurst exponent of surfaces from numerical models (squares) and the Limestone sample (cross). Data points show averages for groups of surfaces generated under the same stress conditions. Error bars on the limestone data shows anisotropy of the JRC. The dashed line shows the relation proposed by Ficker (2017), Eq. 22.

their fractal dimension $D$ calculated from the Hurst exponent as $D = 2 - H$ (Fig. 17). Interestingly, the data from the sandstone sample plots even further below the relation by Ficker (2017). The data from the limestone sample is difficult to compare due to the large anisotropy of the JRC and is therefore not plotted in Fig. 17.

It has been suggested by Ponson et al. (2007) that the observed Hurst exponent is an indicator for the failure mode, $H \approx 0.8$ for "damage fracture", i.e coalescence from micro-cracks and $H \approx 0.4$ for "brittle fracture", i.e. continuous propagation of the crack. However, we have not been able to confirm this for our numerical experiments. Looking at the relative timing of bonds breaking suggests that the fracture surfaces in the DEM models grow by coalescence of micro-cracks despite having a Hurst exponent closer to 0.4. For examples of the general evolution of the micro-crack distribution see Figures S7 and S8 in the

supplement.

The dependence of the variability of the measured Hurst exponent on the size of the analyzed surface on both limestone and sandstone samples suggests the large scatter observed in the Hurst exponents from the numerical models could be a resolution issue. The sandstone sample has a maximum grain size of about $200 \mu m$. This results in a ratio between the length and width of the analyzed fracture surface and the maximum grain size of between 250:1 and 350:1, whereas this ratio is only in the range

between 30:1 and 60:1 in the numerical models. The limestone sample is even more fine-grained than the sandstone sample.

Amitrano and Schmittbuhl (2002) find a weak decrease of the roughness exponent with increasing confinement if no further shear displacement is imposed on the surfaces after fracture. This is similar to the trend observed in our numerical simulation data (Fig. 11), although at different absolute values of the Hurst exponent, which are in the range between 0.3 to 0.45 in our data and between 0.7 to 0.77 in (Amitrano and Schmittbuhl, 2002). Also, this stress dependence can not be directly compared

because of differences in the mechanical properties between the simulated material in our case and the real granite. Amitrano and Schmittbuhl (2002) do not explicitly give the unconfined compressive strength (UCS) of the granite. Extrapolating from

their Figure 3 suggests a value of around 300MPa, although a calculation from their internal cohesion (37MPa) and friction angle ($55 \pm 2$ degrees) gives a value closer to 240MPa. Combined with the confining stress used in their work of $\sigma_3 \approx 20 - 80$MPa this suggests that the ratio between UCS and the confining stress is in a similar range as in the numerical models used here where UCS=80MPa and $\sigma_3 = 0 - 15$MPa.

## 6   Conclusions

Synthetic fracture surfaces have been generated in numerical simulations of rock deformation experiments using the Discrete Element Method (DEM). Results of a statistical analysis demonstrate that the generated surfaces are self-affine. Further analysis has shown no dependency of roughness measures such as RMS roughness and the Joint Roughness Coefficient (JRC) on the confining stress. One exception is the observation that samples fractured under true anisotropic conditions ($\sigma_1 > \sigma_2 > \sigma_3$) show lower JRC and lower RMS roughness than samples fractured under transversal isotropic confinement ($\sigma_1 > \sigma_2 = \sigma_3$), at least for numerical models. For natural rock samples this effect has not been tested yet. Photogrammetric analysis of shear fracture surfaces on two rock samples has shown that the choice of sampling area can influence the roughness data obtained. Results show, for example a variation of $\pm 0.1$ in the Hurst exponent between small sampling areas on the same surface of a rock sample.

Comparing the numerical results with laboratory experiments and additional data obtained from the literature suggests that the trends observed in the numerical parameter study are valid, but it also shows some discrepancies in the absolute values of some of the roughness parameters. In particular, the fracture surfaces generated in the DEM simulations show a higher Joint Roughness Coefficient compared to natural rock samples and a lower Hurst exponent. The comparison also shows a stronger directional anisotropy of the roughness in the real rock samples compared to the numerical simulations. The reason for this result is not clear so far and should be subject to further investigation. One possible cause might be the occurrence of grain size reduction in real rocks, which is not implemented in the current numerical models.

*Author contributions.*  SA performed the numerical simulations, the data analysis on the numerical models and wrote the initial draft of the manuscript, HD performed the analysis of the laboratory samples and edited the manuscript.

*Competing interests.*  The authors declare that they have no competing interests.

*Acknowledgements.*  The work was carried out within the project "PERMEA", funded by the German Federal Ministry of Education and Research under the funding id FKZ 03G0865B. The the triaxial deformation tests were carried out at Technische Universität Darmstadt.

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
