# Peer review of "Roughness of Fracture Surfaces in Numerical Models and Laboratory Experiments"

_Solid Earth, 2021_

## Referee Comment (RC1)

This manuscript explores the possibility of using numerical modelling to hint the morphology of fractures by comparing them with results from laboratory experiments. The modeling is accomplished using DEM methods able to recreate a reliable shape of fracture surfaces, then compared to natural fracture surface from lab experiments. The Authors claim to have explored this comparison in the various stress conditions, yet due to the small size of the generated natural fractures, this comparison is restricted to the two experiments developed in tri-axial conditions. This significantly reduces the impact of the manuscript, that may correctly focus on the used comparisons and requires a more focused title and abstract.

Even in the only two compared experiments there are substantial differences in the roughness results between numerical and experimental results. This mostly derives from the confusion between extensional fractures and faults (i.e. shear fractures). The experimental fractures show a clear shear-related formation (their reported angle to the stress conditions is exactly what expected for faulting). And this might well be the main factor that produces the found roughness differences! The slip along the surface, even if very limited, is responsible for a smoothing process (that, in the case of large faults would derive in to the well-known "fault mirrors", characterized by the almost absence of roughness along the slip direction). This smoothing is proportional to the slip amount, yet could be easily replicated in the numerical modelling (that are intentionally by the author halted at the rupture point, even if the slip starts immediately DURING the enucleation of the shear-fracture surface). This additional behavior can be accomplished by adding the proper directional smoothing kernel on the re-oriented rupture plane (on the "height" component?). My suspect is that the proper application of this filter would significantly reduce the difference in the roughness between numerical modelling and experimental results.

On the other hand, more information are required on the grain-size of the DEM modelling. We know that the efficiency of a stress to break grains is inversely proportional to the grain dimension. That is, the grain-size distribution used in the DEM may affect the roughness of the results. My suggestion to the authors is to include this factor in the modelling and to explore its role on the roughness in the numerical experiments.

On last point is about the discarded fractures related to their small dimensions... I did not fully understand it. Was it a problem of resolution in the scanning with respect to the particle dimensions of the DEM (i.e. to have comparable number of points)? What about improving the scanning resolution or enlarging the samples in the experiments? It would be of great interest to fully explore the generation of real, small extensional fractures.

In conclusion, the manuscript in its present form is not suitable for publication, despite the interesting, and surely inspiring, subject the Authors propose. In this way, I strongly recommend to improve it and resubmit, hoping that my suggestions will be useful.

Here is a list of observations on the text:

Line 6 and forward- why using the term self-affine with respect to self-similar (e.g. Turcotte, 1992, a reference that is strangely missing)

Line 30 it is better to use resolution rather than "scale"

Line 53 Please indicate explicitly the dimensions of the model (3D re-projected on a 2D+height dimension?)

Line 68 cylindrical since is a symmetrical 2D model? This is not clear

Line 87 why 30 degrees? Is it related to the frictional angle, as suggested in: (e.g.) A Nur, H Ron, O Scotti, 1986?

Line 90 more info on this "final step" will ease the understanding of the numerical process

Line 106, 126 and in the manuscript: there is some confusion among the terms "height", "y", and "z". It seems they might indicate the same coordinate axis. Please explicit or indicate the differences

Line 153 please add indications on how the random selection is accomplished

Line 157: it would be useful if you indicate here the relation between the Hurst exponent and the fractal dimension D

Line 163. Here is where a full description of the particle size distribution in the numerical experiment should be added

Line 163 : "a large number"

Line 165 : s1 > s2 > s3 >0

Line 179: please indicate which is the "chosen speed" and why it was chosen since this will affect the rupture roughness

Line 186: please quantitatively specify "rest" and indicate how it has been chosen

Line 195-198 the quantitative comparison between experiments at different speed would prove this sentence

Line 202 among

Line 218-221 Please rephrase Abstract and Introduction accordingly!

Line 226: Apologize for my ignorance: what is mio unit?

Line 224-227 To complete the indications, the distance between lens and object must be specify, since this influence the resolution

Line 240 were calculated, were

Line 264-267 as discussed earlier, the found angle is obvious: faulting (shear plane)

Fig.9 indicate the unit of the x-axis "Distance" scale and its relation to particle size

Line 281-287 as discussed earlier, this is something that has to be fully discussed earlier for its general meaning

Line 304-307 a figure is require to prove it

Line 315-317 is this statistically meaningful?

Line 318-322 Does this mean that their distribution is not self-similar ("self-affine")? A comment will help

Line 339-344 refer to what presented earlier

Line 347-356 According to this sentence, the manuscript should be reshaped to avoid creating disconfirmed expectations by the readers

Line 357-360 the use of two different measure (% and ration) might be confusing for the readers

Line 364-380 refer to what discussed earlier

Line 367-369 add a comment on grinding and smoothing produced by sliding

Line 403 apologize again for my ignorance: what is "Brasilian test"? Maybe some description of it for "the rest of us" will improve the reader comprehension

Line 409-411 I strongly doubt this, according to the relation with the role played by grain-size distribution (see earlier). Is this (forced) sentence necessary? Or better justify it!

Line 415-416 at least the values of the results must be mentioned. As it is written, it seems a subjective elimination of undesired data...

Line 422-435 again the problematics connected to grain-size distribution. My suggestion is to discuss this earlier in the text (in the introduction?)

Line 429 The experiment velocity (i.e. velocity of propagation of fracture surfaces) will influence the reorganization of stress and the influence of local secondary stress components, and therefore the resulting roughness. This has to be taken into account when presenting results. Perhaps a series of experiments at different velocity will highlight this very interesting point.

Line 443 please specify that they result from tri-axial conditions! (and therefore the development of faults - shear surfaces)

Line 451 better "result" rather than "phenomenon" ?

Francesco Salvini

---

## Referee Comment (RC2)

Abe & Deckert produce fracture surfaces in DEM models and laboratory experiments on sandstone and limestone. They measure the roughness of these surfaces with a variety of metrics. They find an insignificant influence of confining stress on fracture roughness, and self-affine numerical and laboratory fractures. One key finding is that biaxial loading produces greater roughness than triaxial loading. However, the manuscript does not adequately explain why this difference in loading would produce the difference in roughness. Throughout the manuscript, there are a few other interesting results that would benefit from additional mechanical explanation. For example, comment #9: why do the natural rock samples have different ranges of roughness in the slip-parallel and -perpendicular directions?

I suggest that this manuscript may be published after the authors provide more thorough explanations of their results. More detailed comments follow below.

Best,
Jess McBeck

1) Line 115: "three of the 47 equations presented there have been chosen". Why did you select these equations?
2) Line 202: "These parameters do not provide a direct match to the mechanical properties of the rocks used in the laboratory tests (Section 3.2), but the important ratio between failure strength of the material and the confining stress applied in the laboratory experiments lies well within the range covered by the numerical models (Fig. 5b)." Was there a particular reason that calibration of the models did not produce the precise values of the Young's modulus and UCS of the lab? Or was the problem that the lab USC ranged from 285 MPa to 80 MPa? Also, the Young's modulus of the lab rocks does not seem to be mentioned in section 3.2. And why is the ratio between the failure strength and confining stress more important than the absolute values of the UCS and Young's modulus? I would imagine that the stiffness of the rock has an impact on the way the rock fractures, and the resulting fracture roughness.
3) Line 224: "3D point cloud data with c. 2.2 mio data points". Presumably mio indicates million here? It could be good to change this abbreviation.
4) Line 222: "Only in one experiment with a confining pressure of 30 MPa post-deformation fragments were large enough for our planned fracture surface analyses (Fig. 4b). " If I understand correctly, the lab experiments yield only two fracture surfaces for which you can calculate the roughness, one from a sandstone and one from a carbonate. If this is the case, then please state it explicitly. Otherwise, it could seem that you have several natural fracture surfaces (i.e., Figure 5).
5) Paragraph at line 237: Why were these estimates outside the range of acceptable values? Could it be related to the inherent roughness of the particles? I wonder if you ran simulations with a larger (0.1-1.0) or smaller (0.3-1.0) particle radii range you would see differences in the calculated roughness, with lower roughness for the larger particle radii range? Later, you show the difference between the height-height correlation functions of a numerical fracture and an arbitrary cut through the model, but did you do a similar calculation for the roughness metrics described in this earlier paragraph? It could be useful to see how the roughness metrics change when you change the particle size distribution, although this would require changing some of the other model parameters, such as the timestep size etc.
6) Line 259: "is possibly at least in part an artifact of the different size of the fracture surfaces between the two model groups" Why is it the case that the size of the fracture surface correlates with the RMS such that larger fracture surfaces would have larger RMS? I understand your explanation of how the size of the fractures differs between the unconfined extension and compression cases, but I do not see why larger fracture surfaces necessarily lead to larger RMS. One may expect greater variations from the mean from a smaller surface as there are less data points producing the mean.
7) Line 319: "The results show that the estimated JRC is dependent on the sampling resolution, i.e. the number of sampling points on the profile, specifically that the calculated value of the JRC is increasing with smaller sampling intervals". Why did you not observe this influence for the numerical models when you calculated the roughness parameters at parallel and perpendicular profiles? And if this is a known effect, then why not shorten the profiles along the dip direction so that they match the length along the strike direction? It would be

interesting to know/constrain if the observed anisotropy is real or not. For example, see the next comment.

8) Line 246 "The results did show that the mean estimated JRCs for the profiles differs by less than 10% between the two direction, which is generally less than the standard deviation between the profiles within one direction" and Figure 15: The laboratory data shows some anisotropy in the roughness, whereas the numerical models do not. Why is this the case? In the discussion, you mention how the indestructible particles and lack of breakable grains could contribute to this point. I also wonder if the faults in the experiments experienced slightly more shear displacement than the fractures in the numerical models. In the numerical models, it's straightforward to select the timestep when the fracture first breaks, and before it slides. But in the lab experiments, I imagine that the deformation could not be stopped at the exact moment of fracture, and that some (perhaps small amount of) slip must occur after the fracture forms (at least for the experiments with compressive loading conditions). This slip could then help produce the anisotropy of the natural lab fractures. In addition, could you also measure the roughness of the numerical fractures after they slip, i.e. some timesteps after the fracture forms? It would be nice to see how the roughness evolves with slip. I wonder if the influence of confining stress on roughness would be larger (not insignificant) when the fracture surfaces slip.

9) Line 333: "Therefore the best estimates for the average JRC of the fracture surfaces produced in the laboratory experiments are for the sandstone J RC ≈ 9 − 11 in the direction parallel to shortening direction in the deformation experiment and J RC ≈ 11 − 13 perpendicular to it (Fig. 15). For the limestone the estimates are J RC ≈ 6.5 − 7.5 in the parallel direction and J RC ≈ 16 − 17 in the perpendicular direction. " Why is the limestone rougher than the sandstone in the slip-perpendicular direction, and less rough in the slip-parallel direction? It's hard to see how this result is directly tied to grain size. Maybe the calcite grains in the limestone are more mechanically anisotropic in strength than the sandstone quartz grains?

10) Line 375: "Based on the results from the numerical models there appears to be a trend towards higher roughness for fracture surfaces generated under transversely isotropic stress conditions, i.e. standard triaxial compression ($\sigma_1 > \sigma_2 = \sigma_3$) compared to those generated under true triaxial conditions ($\sigma_2 \neq \sigma_3$). " Why is this the case? Maybe the orientation of the fracture with respect to the intermediate and minimum compressive stresses could provide insights? How was the fracture oriented relative to the horizontal axes (sigma_2 and/or sigma_3)? Why do smoother surfaces form under true triaxial conditions?

11) Line 419: "Looking at the relative timing of bonds breaking suggests that the fracture surfaces in the DEM models grow by coalescence of micro-cracks despite having a Hurst exponent closer to 0.4." Although the focus of this paper is the fracture surface, it would be useful to see how the fractures coalesce to form this plane, i.e., the position of the fractures (broken bonds) in the timesteps leading the final failure. I would also be interested to see the timeseries of fracture positions in models with true triaxial loading and biaxial loading conditions.

---

## Author Comment (AC1)

Reply to Reviewers Comments

Text in *italics* are reviewers comment, normal text is our reply. The reference section only contains those references which are newly introduced into this reply, i.e. which are not already in the manuscript.

Reviewer 1

*This manuscript explores the possibility of using numerical modelling to hint the morphology of fractures by comparing them with results from laboratory experiments. The modeling is accomplished using DEM methods able to recreate a reliable shape of fracture surfaces, then compared to natural fracture surface from lab experiments. The Authors claim to have explored this comparison in the various stress conditions, yet due to the small size of the generated natural fractures, this comparison is restricted to the two experiments developed in tri-axial conditions. This significantly reduces the impact of the manuscript, that may correctly focus on the used comparisons and requires a more focused title and abstract.*

*Even in the only two compared experiments there are substantial differences in the roughness results between numerical and experimental results. This mostly derives from the confusion between extensional fractures and faults (i.e. shear fractures). The experimental fractures show a clear shear-related formation (their reported angle to the stress conditions is exactly what expected for faulting).*

The fact that the experimental fractures are formed under mode 2 (shear) conditions is the consequence of the applied stress conditions. However, this applies also to the majority of the numerical models, except for one set of unconfined direct tension tests as described in section 3.1., lines 163-167. As a result the angle of the fractures relative to the stress orientations is very similar between the numerical fractures (Fig 1b, Fig 4a) and the experimental ones (Fig 4b,c).

*And this might well be the main factor that produces the found roughness differences!*

We did consider the possibility that the difference in the observed roughness between the numerical and the experimental samples is due to differences in the conditions under which the fractures formed. However, we do not believe that those differences are caused by additional shear motion on the fault in the laboratory experiments compared to the numerical models for several reasons:
First is that the available literature suggests that the small shear displacement occurring in the triax test has only a marginal effect on the roughness, i.e the change in the Hurst exponent H is less than 0.1 [Armitrano & Schmittbuhl 2002, Fig. 12], and that at low confinement H actually goes down with shear. This suggests that the Hurst exponent determined for the lab samples may differ from the value which would have been obtained at zero shear offset some very small amount, but that is clearly not sufficient to explain the difference between lab and numerical experiments.
A second reason is that, on close inspection, the grooves in the limestone sample (Fig 4b) are clearly not caused by abrasion due to the shear offset which might have occurred after fracture. In particular the size of the grooves (several mm deep, ~1cm wide) is too large to be caused by the slip which might have occurred on the surface after the initial failure, which is, specifically under our experimental conditions, at most a few mm (less than 2mm).

The third reason is that we did run a small set of simulations which we did not stop immediately after the formation of a through-going fracture be where we instead continued to deformation until an axial strain of ~12% was achieved and took snapshots of the fracture surfaces at regular data. While it is not clear how much shear offset the experimental fracture surfaces actually did experience, it was clearly much less than the ~13mm of axial displacement which, given the model length of 110mm, would be equivalent to the 12% axial strain in the numerical models.

The obtained Hurst exponents did show no significant trend with increasing strain of the model and offset of the shear fracture. Of the 3 models, and therefore 6 fracture surfaces, 3 showed no significant change of the Hurst exponent at all, i.e. H stayed in a range of +-0.02 of its initial value (038L H2, 040L H2 and 043L H1 in the Figure 1 below), two showed an increase of H between 0.05 and 0.1 (040L H1 and 043L H2 in Fig. 1) and one (038L H1) even shows an initial drop in H for small strains, which is not fully recovered by the subsequent increase at larger strains.

[Figure]

Figure 1: Hurst exponent of fracture surfaces vs. axial strain of deformed sample for 6 surfaces from 3 numerical models. In the legend, the 1st part of the description (038L) is the model ID, the 2nd part (H1, H2) distinguished between the two surfaces of the fracture.

While the average of the Hurst exponents from the 6 surfaces (Fig. 2 below) could be considered as showing a slight increasing trend for strains up to ~8%, the increase of 0.03 is about an order of magnitude too small to explain the observed differences between numerical and experimental surfaces. However, it would be compatible with the effect observed by Armitrano & Schmittbuhl [2002].

We did not include those data in the manuscript for two reasons, one being that a systematic study of the relation between shear offset and roughness parameters of fault surfaces was not within the scope of the work, the other being that of course the data shown above are not sufficient to draw any statistically valid conclusions on the details how the Hurst-exponent evolves with shear. However, given that the data does provide a strong suggestion that the observed differences in the Hurst-exponent between numerical

and experimental fractures are not due to possible differences in the amount of shear, they can probably be included in the discussion section of the manuscript. We will therefore add a paragraph discussing this issue to  the manuscript and the Figures  1 and 2, and also Figure 1 from the reply to reviewer 2 (page 5 there) to the supplementary material.

[Figure]

Figure 2: Average of the Hurst exponents of the 6 surfaces shown in Fig. 1 vs. model strain. Error bars show standard deviation.

*The slip along the surface, even if very limited, is responsible for a smoothing process (that, in the case of large faults would derive in to the well-known "fault mirrors", characterized by the almost absence of roughness along the slip direction).*

We agree with the reviewer that slip along the surface can result in a significant reduction of the Hurst-exponent for profiles parallel to the slip direction down to values below 0.5 as, for example, shown in Table 1b in [Candela et al., 2012]. However, those large reductions appear to apply mainly to faults with large amounts of slip, i.e. several meters up to kilometers. Data available in the Literature [Amitrano & Schmittbuhl 2002, Davidesko et al., 2014, Badt et al. 2016] suggests that at least under brittle conditions and for shear amounts less that a few cm the Hurst-exponents remain largely unchanged. As mentioned above, shortening in our experimental tests did not exceed 2mm.

*This smoothing is proportional to the slip amount, yet could be easily replicated in the numerical modeling (that are intentionally by the author halted at the rupture point, even if the slip starts immediately DURING the enucleation of the shear-fracture surface). This additional behavior can be accomplished by adding the proper directional smoothing kernel on the re-oriented rupture plane (on the "height" component?). My suspect is that the proper application of this filter would significantly reduce the difference in the roughness between numerical modelling and experimental results.*

It appears that, in particular for small amounts of shear, the statement that "*smoothing is proportional to the slip amount*" may not be generally applicable in all situations. It has in fact been suggested in the literature that the roughness evolution of fault surfaces might actually undergo a roughening stage [Davidesko et al., 2014, section 5.3]. If this is indeed the case, applying an appropriate filter to the surface data to simulate the effect of shear-induced smoothing as a post-processing step would be highly non-trivial.

*On the other hand, more information are required on the grain-size of the DEM modelling. We know that the efficiency of a stress to break grains is inversely proportional to the grain dimension. That is, the grain-size distribution used in the DEM may affect the roughness of the results. My suggestion to the authors is to include this factor in the modelling and to explore its role on the roughness in the numerical experiments.*

We agree with the reviewer that size dependent comminution of grains can clearly have an influence on the evolution of the roughness of fracture surfaces. However, to include this effect into the numerical models would increase the computational cost by at least an order of magnitude if using an empirical model to break individual particles and replace them with a set of smaller particles like the one proposed by Cleary [2001] and probably more if the grain fracture process is actually modeled fully as in Abe & Mair [2005]. An inclusion of grain fracture into the models would therefore result in impractically long computing times, even when using current high performance computing facilities. This would apply in particular to the simulation of statistically useful model ensembles. We therefore did not include intra-grain fracture into the numerical models and we explicitly mention this as one of the differences between the models and real rock (Discussion, lines 365-374, Conclusions, last sentence).

*On last point is about the discarded fractures related to their small dimensions… I did not fully understand it. Was it a problem of resolution in the scanning with respect to the particle dimensions of the DEM (i.e. to have comparable number of points)?*

In our samples we had the difficulty, that the deformed samples disintegrated into small pieces. However, there is a limitation in the size of samples (=planes) which can be analysed. The key problem is that with the available tools for the analysis of the experimental fracture surfaces, i.e. photogrammetry using a standard digital SLR with a macro lens (section 3.2, lines 223-227), the resolution of the obtained surface data is restricted to about 0.1mm. Given that the analysis of the surfaces roughness required at least around 100 sample points per direction (section 4.2, lines 327-330), and the fact that the sampling interval should be somewhat larger than the resolution of the data, this restricts the minimum size the usable surfaces to at least 2-3cm in each dimension of the surface. Unfortunately such large fragments were produced only in very few of the deformation experiments. The photo below shows the typical remains from one of the triaxial experiments.

[Figure]

*Figure 3: Fragments of a limestone sample after a triaxial deformation experiment.*

150     *What about improving the scanning resolution or enlarging the samples in the experiments?*

    While this would theoretically be a possible way to circumvent the problem, it is unfortunately not feasible within the current study due to the lack of the necessary tools
155     and resources to do so. In particular, the dimension of the rock samples (55mm diameter x 110mm length) had to be chosen as standard sample size to fit the available deformation apparatus.

    *It would be of great interest to fully explore the generation of real, small extensional*
160     *fractures.*

    We agree that this would be an interesting topic. However, to do this one would need different techniques for acquiring the necessary height field at the  required resolution. We also must stress that the investigation of real, small extensional fractures was not the topic
165     of our paper.

*Here is a list of observations on the text:*
*Line 6 and forward- why using the term self-affine with respect to self-similar (e.g.*
*Turcotte, 1992, areference that is strangely missing)*

We are using the definition from Mandelbrot [1985] and Bouchaud [1997] defining self-similarity as statistical invariance under isotropic scaling and self-affinity as statistical invariance under an affine transformation. To quote Bouchaud [1997]: "Unlike self-similar objects, self-affine structures, being intrinsically anisotropic, are not statistically invariant through a global dilation but rather through an affine transformation". This should, to our understanding, actually be equivalent to the description of a self-affine fractal on page 8 of [Turcotte 1992]. In regard to the description of rough surfaces Turcotte [1992] actually states that "A cross-section of topography with elevation plotted against position along a linear track is not a self-similar fractal; however, it is usually a self-affine fractal."
The reference to [Turcotte 1992] can of course be added to the relevant section of the manuscript.

*Line 30 it is better to use resolution rather than "scale"*

In the context of that sentence the term "scale" is actually the correct one. The RMS deviation of a surface from a given average plane depends on what scale, for example mm, meters or kilometers, it is measured, not so much at what resolution that deviation is sampled.

*Line 53 Please indicate explicitly the dimensions of the model (3D re-projected on a 2D+height dimension?)*

As there is no "model" mentioned in line 53 we are not completely sure what the reviewer meant here. This section (2.1 Discrete Element Method) only describes the numerical simulation approach as such, not the specific models used with that approach later in the manuscript.

*Line 68 cylindrical since is a symmetrical 2D model? This is not clear*

The assumption of cylindrical bonds between individual DEM particles is a fairly common way to parameterize the brittle-elastic interactions between those particles [Potyondy & Cundall 2004, Weatherly 2011]. The choice of a circular cross section for the bonds is largely due to the fact that the particles are assumed to be spherical, but it is also mathematically convenient.

*Line 87 why 30 degrees? Is it related to the frictional angle, as suggested in: (e.g.) A Nur, H Ron, O Scotti,1986?*

No. This parameter has no physical meaning at all. When developing and testing the algorithm, the 30 degrees separation between the different view directions used to identify surface particles has been found to provide a good balance between computational effort and quality of the reconstructed surfaces. Figure 2 in the manuscript will be changed to make this more clear. See modified figure below.

[Figure]

Figure 2 (modified): Simplified 2D sketch of the ray-casting method. The gray particles are assumed to belong to the same fragment of the deformed sample and the black crosses show the fragment surface calculated from the line-particle intersections. **Black lines and black arrow show primary view direction, light gray lines and arrows show additional view directions at a 30 degree angle to the primary direction.**

*Line 90 more info on this "final step" will ease the understanding of the numerical process*

The final step to determine which other fragment of the model is closest to a given surface particle essentially involves checking all other particles in order of their distance to this particle if they belong the same fragment of the model as the surface particle or to another large fragment of the model until one is found belonging to another fragment. On a more technical level it does involve the use of an acceleration grid to avoid the calculation of all particle-particle distances while still being able to pre-sort the particles roughly in order of their distance to the surface particle for which the calculation is performed.

*Line 106, 126 and in the manuscript: there is some confusion among the terms "height", "y'", and "z'". It seems they might indicate the same coordinate axis. Please explicit or indicate the differences*

"Height" generally means a distance from a given plane measured along the plane normal as described in line 131 and Fig. 3A in the manuscript.

*Line 153 please add indications on how the random selection is accomplished*

The software (python script) written by the authors for the surface analysis uses the function "numpy.random.choice()" from the numpy-library ([www.numpy.org](www.numpy.org)) to draw a random selection from an array of points. Technical details of that function can be found in the online documentation of the numpy-library ([https://numpy.org/doc/](https://numpy.org/doc/)).

*Line 157: it would be useful if you indicate here the relation between the Hurst exponent and the fractal dimension D*

245

The Hurst H exponent and the fractal dimension D of an object are related as D=2-H for a 1D-profile [Mandelbrot 1985] or, more generally, D=n+1-H where n is the dimension of the object [Yang & Lo, 1997], i.e. n=1 for a profile and n=2 for a surface. We will add this explanation in the modified manuscript.

250

*Line 163. Here is where a full description of the particle size distribution in the numerical experiment should be added*

The insertion based particle packing algorithm by Place & Mora [2001] mentioned in line
255 194 does produce particles with a power-law size distribution with an exponent of approximately -3. I.e. the number of particles in a given size bin is proportional to $r^{-3}$. See Figure 3 below for an example.

[Figure]

*Figure 4: Log-log plot of the size distribution of the particles in one of the models.*

260

*Line 163 : "a large number"*
*Line 165 : s1 > s2 > s3 >0*

Will be changed.
265

*Line 179: please indicate which is the "chosen speed" and why it was chosen since this will affect the rupture roughness*

As described in lines 195ff, the loading speed was set to 17cm/s in order to be sufficiently similar to experimental values while keeping the required computational cost at a viable level. Given that this speed is nearly 4 order of magnitude below the dynamic fracture propagation speed, it should not influence the processes happening during rupture propagation to any significant degree.

*Line 186: please quantitatively specify "rest" and indicate how it has been chosen*

We are assuming this concerns line 184, not 186.
The "rest" phase was chosen to be 10000 time steps in the numerical model. This was sufficient for the particle kinetic energy to be reduced by about an order of magnitude.

*Line 195-198 the quantitative comparison between experiments at different speed would prove this sentence*

Due to the sensitivity of the fracturing process to small perturbations, the speed at which a model is deformed can indeed influence the outcome of individual simulations, making the direct comparison of experiments at difference speeds difficult. In order to compare the outcomes of such experiments statistically we therefore performed a set of 6 simulations at a reduced compression speed of ~8.5cm/s, i.e. half of the rate used in the majority of the simulations. All those simulations were performed under transverse isotropic stress conditions, i.e. $\sigma_2=\sigma_3$. The analysis of the 20 fracture surfaces produced in those simulations showed that neither the Hurst-exponent nor the RMS roughness or Joint Roughness Coefficient (JRC) were statistically significantly different from those of surfaces produced with the faster compression rate under the same stress conditions. The average Hurst exponents obtained from the "slow" experiments was H=0.408 with a standard deviation of 0.064 whereas for the "fast" experiments it was 0.375±0.084. The RMS roughness was 2.23±0.45 for the slow vs. 2.39±0.77 for the fast models. The calculated JRC is 28.9±1.9 for the slow and 29.7±2.8 for the fast models.

*Line 202 among*

*Line 218-221 Please rephrase Abstract and Introduction accordingly!*

We are not sure which of the aspects mentioned in those lines the reviewer refers to. We could explicitly mention in the abstract and the introduction that we only had a small number of usable samples from the laboratory experiments available.

*Line 226: Apologize for my ignorance: what is mio unit?*

Millions. This will be changed in a revised manuscript.

*Line 224-227 To complete the indications, the distance between lens and object must be specify, since this influence the resolution*

The distance between object and lens was variable, but usually slightly larger than the minimum focus distance of the lens, which is approximately 5cm, measured from the front lens to the object or, according to lens specifications, 16.3cm if measured from the camera

sensor. I.e. the photos were taken at varying distances between 5cm and 10cm from the front lens to the object.

320 *Line 240 were calculated, were*

Will be changed.

*Line 264-267 as discussed earlier, the found angle is obvious: faulting (shear plane)*
325

The angle was only mentioned here to highlight the difference in orientation, and therefore maximum possible size, between the fractures generated by shear failure due to triaxial compression of the model and those generated by tensile failure due to uniaxial extension of the model.
330

*Fig.9 indicate the unit of the x-axis "Distance" scale and its relation to particle size*

The unit used for the distance scale is the radius of the largest particles in the DEM model.

335 *Line 281-287 as discussed earlier, this is something that has to be fully discussed earlier for its general meaning*

We probably should made this statement more precise to avoid a possible misunderstanding. The sentence "Only the absolute value of the roughness depends
340 somewhat on the size range of the particles." only applies to the "cut" surfaces, not to surfaces obtained by numerical simulation of fracture.
We will change this sentence to "Only the absolute value of the roughness **of the cut surfaces** depends somewhat on the size range of the particles.".

345 *Line 304-307 a figure is require to prove it*

See Figure 5 (Limestone sample) and Figure 6 (Sandstone sample) below.

[Figure]

*Figure 5: Height-height correlation function for the Limestone sample. Top: Full distance range, bottom: range used for fitting a linear relation and calculation of Hurst-exponent.*

[Figure]

*Figure 6: Height-height correlation function for the Sandstone sample. Top: Full distance range, bottom: range used for fitting a linear relation and calculation of Hurst-exponent.*

*Line 315-317 is this statistically meaningful?*

355 As there are only 2 data points where this effect is observed, the statistical significance isn't really clear. The statistical argument that those two points are indeed outliers would be their H-value is more than 2 standard deviations below the closest other value (0.5 vs. 0.67, standard deviation is 0.08). The other argument would be that they are the only data points where we don't see a clear single linear section in the height-height correlation
360 function.

*Line 318-322 Does this mean that their distribution is not self-similar ("self-affine")? A comment will help*

365 On the contrary. As stated in lines 322/323, "It is also to be expected based on the fact that the analyzed surfaces are self-affine.", i.e. the observations described in lines 318-322 are in fact a consequence of the self-affine properties of the surfaces.

*Line 339-344 refer to what presented earlier*
370
We do not fully understand this comment. We refer to what was already presented in Line 341: "The results of the analysis of the simulation data (Section 4.1) shows that..."

*Line 347-356 According to this sentence, the manuscript should be reshaped to avoid*
375 *creating disconfirmed expectations by the readers*

We are not sure what the reviewer means with "disconfirmed expectations".

*Line 357-360 the use of two different measure (% and ration) might be confusing for the*
380 *readers*

We would prefer not to change this sentence.

*Line 364-380 refer to what discussed earlier*
385
As discussed in lines 120-131 of this reply, the inclusion of a grain size reduction mechanism would be interesting for further study, but is currently outside the scope of this work.

390 *Line 367-369 add a comment on grinding and smoothing produced by sliding*

We will add the following sentences: "Smoothing due to abrasion while sliding is, in general, an important mechanism for the modification of rough surfaces. However, experimental data published in the literature [Amitrano & Schmittbuhl 2002, Davidesko et
395 al., 2014, Badt et al. 2016] suggest that it is unlikely to have a sufficiently large effect at the small shear offsets in both numerical models and experimental samples studied here to explain the observed differences. "

*Line 403 apologize again for my ignorance: what is "Brasilian test"? Maybe some*
400 *description of it for "the rest of us" will improve the reader comprehension*

A "Brazilian test", also known as "Brazilian disk test" is a standard geomechanical test to obtain the tensile strength of materials by applying a compressive load to two opposite points on the circumference of a circular disk of the material. This loading results in a

405 tensile stress in the central part of the disk which is oriented perpendicular to the line connecting the two loading points. It is often preferred to a direct tension test because it is easier to perform. However, it only works if the material has a sufficiently high ratio between compressive and tensile strength, otherwise it will fail in shear instead of tension. See, for example Jaeger et al., [2007] or Fjaer et al. [2008].

410 An appropriate reference will be added.

*Line 409-411 I strongly doubt this, according to the relation with the role played by grain-size distribution (see earlier). Is this (forced) sentence necessary? Or better justify it!*

415 In order to clarify that this sentence is meant to be understood in the context of the aforementioned observations by Nigon et al. [2017], we will modify it to "This difference in scales shows that the Hurst-exponents in our numerical models are completely calculated above the "transition scale" of Nigon et al. [2017] and therefore should belong to the regime described as "jointing induced roughness" by them. This means that the low values

420 of the Hurst-exponents in the numerical can not be explained by the "grain induced roughness" regime of Nigon et al. [2017]."

*Line 415-416 at least the values of the results must be mentioned. As it is written, it seems a subjective elimination of undesired data…*

425

The relation between Hurst-exponents and JRC for the limestone sample could, in principle, be analyzed separately for the directions parallel and perpendicular to the shortening direction to take the strong anisotropy of the JRC into account. If this was done, the data for the shortening-parallel direction (H~0.55, i.e. D-1=0.45, JRC~6.5...7.5) would

430 plot in Fig. 17 even further below the relation proposed by Ficker than the sandstone, whereas the data in the perpendicular direction (H~0.605, D-1~0.395, JRC~16...17) would be closer, but still below.

*Line 422-435 again the problematics connected to grain-size distribution. My suggestion is*
435 *to discuss this earlier in the text (in the introduction?)*

The data (Fig. 14 in the manuscript) suggests that the variability of the Hurst-exponents is likely be influenced by the "resolution", i.e. the ratio between grain size and surface extension. However, the mean value of the Hurst-exponents doesn't seem to be

440 influenced. In addition, a small number of tests using numerical models with a different particle size distribution, specifically a larger range of particle radii has also shown no influence of the particle size distribution on the Hurst-exponent in these models. See also our response to Reviewer 2, lines 105ff.

It is of course possible that the grain size distribution might have an effect if grain fracture
445 is included in the model. However, as explained in lines 368-372 in the manuscript, this would be beyond the scope of the current work.

*Line 429 The experiment velocity (i.e. velocity of propagation of fracture surfaces) will influence the re-organization of stress and the influence of local secondary stress*
450 *components, and therefore the resulting roughness. This has to be taken into account when presenting results. Perhaps a series of experiments at different velocity will highlight this very interesting point.*

Not shure how this relates to line 429, but care should be taken not to confuse the loading
455 velocity, i.e. the velocity with which the deformation is applied to the top and bottom surface of the sample in the numerical experiments, with the fracture propagation velocity.

The first on is a boundary condition, which can be controlled by changing a parameter in the simulation, and is set to ~17cm/s in the simulations presented here. The latter is a result of the internal dynamics of the simulated material and is purely controlled by its elastic parameters, i.e. shear and compressive modulus and density.

It would of course be possible to change the fracture propagation velocity in the models by varying the elastic properties of the simulated material, but given that the fracture roughness data published in the literature seem to be largely independent of rock type, and therefore rock elastic properties, it is not clear whether this would actually be a promising direction.

*Line 443 please specify that they result from tri-axial conditions! (and therefore the development of faults -shear surfaces)*

We will change "...Photogrammetric analysis of fracture surfaces" to "...Photogrammetric analysis of shear fracture surfaces"

*Line 451 better "result" rather than "phenomenon" ?*

Will be changed.

References:

Badt, N., Hatzor, Y. H., Toussaint, R. and A. Sagy, 2016, Geometrical evolution of interlocked rough slip surfaces: The role of normal stress, Earth and Planetary Science Letters 443, 153–161

Cleary, P.W., 2001. Recent advances in DEM modelling of tumbling mills. Miner. Eng. 14, 1295–1319

Davidesko, G., Sagy, A. and Y. H. Hatzor, 2014, Evolution of slip surface roughness through shear, Geophys. Res. Lett., 41, 1492–1498, doi:10.1002/2013GL058913

Fjaer, E., Holt, R.M., Horsrud, P., Raaen, A.M. and R. Risnes, 2008, Petroleum related Rock mechanics, 2nd edition, Elsevier

Jaeger, J.C. Cook, N.G.W.  and R.W. Zimmerman, 2007, Fundamentals of rock mechanics, 4th edition, Blackwell

Potyondy, D.O. and P.A. Cundall, 2004, A bonded-particle model for rock, Int. J. Rock Mech. Min. Sci., 41, 1329-1364

Turcotte, D.L., Fractals, chaos, self-organized citicality and tectonics, 1992, Terra Nova, 4, 4-12

Yang, Z.Y. and S.C. Lo, 1997, An Index for Describing the Anisotropy of Joint Surfaces, Int. J. Rock Mech. Min. Sci., 34(6), 1031-1044

---

## Author Comment (AC2)

**Reply to Reviewers Comments**

Text in *italics* are reviewers comment, normal text is our reply. The reference section only contains those references which are newly introduced into this reply, i.e. which are not already in the manuscript.

**Reviewer 2**

5

Abe & Deckert produce fracture surfaces in DEM models and laboratory experiments on sandstone and limestone. They measure the roughness of these surfaces with a variety of metrics. They find an insignificant influence of confining stress on fracture roughness, and self-affine numerical and laboratory fractures. One key finding is that biaxial loading produces greater roughness than triaxial loading. However, the manuscript does not adequately explain why this difference in loading would produce the difference in

15 roughness. Throughout the manuscript, there are a few other interesting results that would benefit from additional mechanical explanation. For example, comment #9: why do the natural rock samples have different ranges of roughness in the slip-parallel and perpendicular directions?

*I* suggest that this manuscript may be published after the authors provide more thorough explanations of their results. More detailed comments follow below.

- Best, Jess McBeck
- 25 1) Line 115: "three of the 47 equations presented there have been chosen". Why did you select these equations?

We did choose these 3 equations because they all show a high correlation coefficient, 0.986 for JRC1 (Eq. 5), 0.951 for JRC31 (Eq. 6) and 0.971 for JRC34 (Eq. 7) [Li & Zhang,

- 30 Table 2] and because we wanted to use 3 equations based on different underlying roughness measures. JRC1 is based on the Root mean square of the first deviation of the profile as defined in [Li and Zhang, 2015, Table 1], JRC31 on the Roughness profile index and JRC34 on the Profile elongation index.
- 35 2) Line 202: "These parameters do not provide a direct match to the mechanical properties of the rocks used in the laboratory tests (Section 3.2), but the important ratio between failure strength of the material and the confining stress applied in the laboratory experiments lies well within the range covered by the numerical models (Fig. 5b)." Was there a particular reason that calibration of the models did not produce the precise values
- 40 of the Young's modulus and UCS of the lab? Or was the problem that the lab USC ranged from 285 MPa to 80 Mpa?

It would have technically been possible to calibrate the DEM material to a UCS of 285MPa. The reason that this wasn't done is that, due to factors outside our control, the laboratory experiments were performed after the bulk of the numerical modeling had been done. So the DEM material was calibrated to typical values for the rock types used as described in the literature. Unfortunately it turned out that the limestone has an unusually high strength.

50 Also, the Young's modulus of the lab rocks does not seem to be mentioned in section 3.2.

The missing values (48GPa for the limestone, 12.5GPa for the sandstone) will be added.

- And why is the ratio between the failure strength and confining stress more important than 55 the absolute values of the UCS and Young's modulus? I would imagine that the stiffness of the rock has an impact on the way the rock fractures, and the resulting fracture roughness.
- We would argue that indeed the ratio between strength and stress is the more important 60 factor. The reason is that fracture criteria can typically be expressed in a non-dimensional way if they are formulated in terms of stress-strength ratios and that many aspects of fracturing are determined by where on a failure envelope the stress state is located under which a particular fracture is formed.
- However, we would also agree that rock stiffness will have an influence on the way the rock fractures. This applies in particular because the ratio between stiffness and strength 65 will determine the failure strain. The unconfined failure strains of the calibrated numerical models and the rock samples differ by only a factor of 2, i.e. the DEM material fails at a strain of ~3\*10-3 under uniaxial compression, both rock types at about 6\*10-3. We did run a few models with higher strength during the calibration phase which did not show any
- 70 obvious differences.

3) Line 224: "3D point cloud data with c. 2.2 mio data points". Presumably mio indicates million here? It could be good to change this abbreviation.

75 Will be changed.

> 4) Line 222: "Only in one experiment with a confining pressure of 30 MPa postdeformation fragments were large enough for our planned fracture surface analyses (Fig. 4b). " If I understand correctly, the lab experiments yield only two fracture surfaces for

which you can calculate the roughness, one from a sandstone and one from a carbonate. 80 If this is the case, We did then please state it explicitly. Otherwise, it could seem that you have several natural fracture surfaces (i.e., Figure 5).

We will modify Figure 5b to make it clear that only two of the laboratory tests were used. 85 See panel (b) in figure below.

Figure 5 (modified)

5) Paragraph at line 237: Why were these estimates outside the range of acceptable values? Could it be related to the inherent roughness of the particles?

There is a contribution of the intrinsic particle roughness to the JRC, but as discussed in lines 346-356 it is too small to explain the very high JRC values calculated for the numerical models.

95

I wonder if you ran simulations with a larger (0.1-1.0) or smaller (0.3-1.0) particle radii range you would see differences in the calculated roughness, with lower roughness for the larger particle radii range? Later, you show the difference between the height-height correlation functions of a numerical fracture and an arbitrary cut through the model, but did

- 100 you do a similar calculation for the roughness metrics described in this earlier paragraph? It could be useful to see how the roughness metrics change when you change the particle size distribution, although this would require changing some of the other model parameters, such as the timestep size etc.
- 105 We did not run models using a narrower range particle sizes, such as 0.3-1.0. However, we did run 2 small sets of simulations with a wider range of particle sizes (0.15-1.0 and 0.1-1.0). The sets consisted of 5 simulations each, all performed under true triaxial conditions using  $\sigma_2$ =6MPa and  $\sigma_3$ =0. Results did not show a statistically significant difference in Hurst exponent or JRC compared to the equivalent simulations performed using a particle radius range of 0.2-1.0. See table below.

| particle size range | Hurst exponent | JRC      | JRC anisotropy |
|---------------------|----------------|----------|----------------|
| 0.2-1.0             | 0.414±0.5      | 26.1±1.8 | 2.0%           |
| 0.15-1.0            | 0.415±0.85     | 25.4±2.4 | 3.2%           |
| 0.1-1.0             | 0.398±0.96     | 24.2±2.3 | 0.4%           |

We will add this data and the explanation above to the discussion section of the manuscript.

- 6) Line 259: "is possibly at least in part an artifact of the different size of the fracture surfaces between the two model groups" Why is it the case that the size of the fracture surface correlates with the RMS such that larger fracture surfaces would have larger RMS? I understand your explanation of how the size of the fractures differs between the unconfined extension and compression cases, but I do not see why larger fracture surfaces necessarily lead to larger RMS. One may expect greater variations from the
- mean from a smaller surface as there are less data points producing the mean.

Given that the fracture surfaces are self-affine, it would be expected that the variation in "height" between points on the surface scales with their distance as shown in Fig. 9 in the manuscript. Therefore a larger surface would be expected to contain larger height differences, resulting in a larger RMS roughness.

7) Line 319: "The results show that the estimated JRC is dependent on the sampling resolution, i.e. the number of sampling points on the profile, specifically that the calculated value of the JRC is increasing with smaller sampling intervals". Why did you not observe this influence for the numerical models when you calculated the roughness parameters at parallel and perpendicular profiles?

The JRC data for the numerical models are calculated using a constant sampling interval and therefore a variable number of sampling points depending on the profile length. (see also next comment below).

And if this is a known effect, then why not shorten the profiles along the dip direction so that they match the length along the strike direction? It would be interesting to know/constrain if the observed anisotropy is real or not. For example, see the next comment.

140

The roughness measures underlying the JRC calculations ("Root mean square of the first deviation of the profile", Eq. 9 in the manuscript) are local measures which depend only on the differences between neighboring sample points. The calculated JRC values do scale with the size of the sampling interval, i.e. the distance between sampling points, not the total number of samples. So "shortening" of the dip-direction profiles in the sense of clipping them to the same length and number of sample points as the strike-direction profiles might change the JRC of individual profiles, but it does not affect the overall statistics.

Additionally, at least for the limestone the observed anisotropy is much too large to be explained by sampling effects. The ratio between the JRC calculated for dip- and strikedirection profile using the same sample point distance differs by a factor of at least 2 for all sample point distances tested (Fig. 15 in the manuscript), which is significantly larger than the observed change of the JRC for difference sample intervals (same Figure).

8) Line 246 "The results did show that the mean estimated JRCs for the profiles differs by less than 10% between the two direction, which is generally less than the standard deviation between the profiles within one direction" and Figure 15: The laboratory data

- 160 shows some anisotropy in the roughness, whereas the numerical models do not. Why is this the case? In the discussion, you mention how the indestructible particles and lack of breakable grains could contribute to this point. I also wonder if the faults in the experiments experienced slightly more shear displacement than the fractures in the numerical models. In the numerical models, it's straightforward to select the timestep when
- 165 the fracture first breaks, and before it slides. But in the lab experiments, I imagine that the deformation could not be stopped at the exact moment of fracture, and that some (perhaps small amount of) slip must occur after the fracture forms (at least for the experiments with compressive loading conditions). This slip could then help produce the anisotropy of the natural lab fractures. In addition, could you also measure the roughness
- 170 of the numerical fractures after they slip, i.e. some timesteps after the fracture forms? It would be nice to see how the roughness evolves with slip. I wonder if the influence of confining stress on roughness would be larger (not insignificant) when the fracture surfaces slip.
- 175 As stated in the reply to reviewer 1 (page 1, lines 46-51), the grooves in the limestone sample, which are largely responsible for the high anisotropy measured there, do not appear to be abrasion features.

To look at the roughness evolution of the fracture surfaces with increasing deformation of the sample we did perform a small number of simulations which did not stop immediately

180 after the formation of the fractures, but instead continued deformation to a total axial strain of up to 12%. This is significantly larger than the strain occurring in the laboratory experiments, where total axial shortening did not exceed about 2%. In particular, the amount of shortening occurring after the peak axial stress was reached, i.e. after failure, was generally less than 1%.

- For the surfaces extracted from those simulations, we did not observe a significant change of the Hurst-exponent with increasing shear offset of the surfaces (see reply to reviewer 1, pages 1-3). For one of the models we did also calculate the JRC of the surfaces at various stages of the simulation (see Figure below). The data shows that there is also no significant change of the JRC for the shear offset considered in this model, which would be equivalent to ~1cm in the laboratory samples, and under the conditions of this model (true
- triaxial stress,  $\sigma_2$ =7.5MPa,  $\sigma_3$ =3MPa). This seems to confirm again that under the small shear offsets relevant for our experiments, there is very little evolution of the surface roughness, at least as far as it concerns the roughness parameters calculated here (Hurst-exponent, JRC). In particular,
- 195 the data would suggest that any effects due to small, but the non-zero, shear offset in the laboratory experiments are much too small to explain the observed differences between numerical simulations and laboratory experiments. We will add a paragraph discussing this issue to the "Discussion" section of the manuscript

200

and the figure below, and also Figures 1 and 2 from the reply to reviewer 1 (pages 2,3) to the supplementary material.

Figure 1: Evolution of the JRC with increasing shear offset of the surfaces. Data points are averages of the two surfaces of the same shear fracture.

9) Line 333: "Therefore the best estimates for the average JRC of the fracture surfaces produced in the laboratory experiments are for the sandstone JRC ≈ 9 - 11 in the direction parallel to shortening direction in the deformation experiment and JRC ≈ 11 - 13 perpendicular to it (Fig. 15). For the limestone the estimates are JRC ≈ 6.5 - 7.5 in the

parallel direction and JRC  $\approx$  16 – 17 in the perpendicular direction. "Why is the limestone rougher than the sandstone in the slip-perpendicular direction, and less rough in the slipparallel direction?

210

The high roughness in the perpendicular direction in the limestone sample is mainly due to the large longitudinal grooves on the surface visible in Fig 4b and Fig 13a. The mechanism responsible for the generation of those structures is, unfortunately, unclear to us.

It's hard to see how this result is directly tied to grain size. Maybe the calcite grains in the 215 limestone are more mechanically anisotropic in strength than the sandstone quartz grains?

This is an interesting idea which could probably be tested in future work. However, we do currently have no indication that the calcite grains in the limestone show any sort of 220 preferential alignment, which, one would presume, should be necessary for the grain scale strength anisotropy to have any effect much beyond the grain scale. Unfortunately all limestone samples were taken in the same orientation with the long axis of the cylindrical samples, and therefore  $\sigma_1$  during the triaxial tests, perpendicular to the bedding of the

- 225 limestone, so we do not know if the macroscopic strength of the limestone is anisotropic or not. However, SEM images of limestones sampled from the same location appear not to show any preferred grain orientation.
- 10) Line 375: "Based on the results from the numerical models there appears to be a trend 230 towards higher roughness for fracture surfaces generated under transversely isotropic stress conditions, i.e. standard triaxial compression ( $\sigma 1 > \sigma 2 = \sigma 3$ ) compared to those generated under true triaxial conditions ( $\sigma 2 \neq \sigma 3$ ). "Why is this the case? Maybe the orientation of the fracture with respect to the intermediate and minimum compressive stresses could provide insights? How was the fracture oriented relative to the horizontal

235 axes (sigma 2 and/or sigma 3)?

> The fracture orientations were as one would expect under the stress conditions. The dip angle was within 25-35° of  $\sigma_1$ , i.e. 55-65° assuming  $\sigma_1$  to be vertical. The strike was usually within ~10° of  $\sigma_2$  in the true triaxial models ( $\sigma_2 = \sigma_3$ ) and more or less randomly distributed in the transverse isotropic conditions ( $\sigma_2 = \sigma_3$ ).

We will add this sentence to the "Results" section of the manuscript.

Why do smoother surfaces form under true triaxial conditions?

245

250

240

Unfortunately we can only speculate about the reasons at this point. One, purely speculative, idea might be that, if we assume that the through-going fractures which we analyze form by coalescence from smaller, precursory, fractures, those precursory fractures have their strike angles constrained to a narrow range if  $\sigma_2 = \sigma_3$ , but that there is no such constraint if  $\sigma_2 = \sigma_3$ . If this is the case, than the coalescence of those fractures might lead to smoother large-scale surfaces if they all have similar orientations. Unfortunately the numerical models used in this work do not have the resolution necessary to study this process.

We will add a paragraph about this to the "Discussion" section of the manuscript. 255

11) Line 419: "Looking at the relative timing of bonds breaking suggests that the fracture surfaces in the DEM models grow by coalescence of micro-cracks despite having a Hurst exponent closer to 0.4." Although the focus of this paper is the fracture surface, it would be useful to see how the fractures coalesce to form this plane, i.e., the position of the fractures (broken bonds) in the timesteps leading the final failure. I would also be interested to see the timeseries of fracture positions in models with true triaxial loading and biaxial loading conditions.

- 265 While the details of the evolving micro-crack distribution prior to the main failure of the sample have indeed been beyond the scope of the work presented in the manuscript, we did store sufficient snapshots from a number of models to reconstruct time series of micro-crack distributions where every "micro-crack" visualized is a single particle-particle bond breaking. See below a set of snapshots from a biaxial ( $\sigma_2 = \sigma_3 = 6$ MPa) and a triaxial ( $\sigma_2$  = 15MPa,  $\sigma_3 = 6$ MPa) model. Timing of the snapshots is shown on a plot of axial stress
- 270 = 15 MPa,  $\sigm

---

## Author Comment (AC3)

**Reply to Reviewers Comments**

Text in *italics* are reviewers comment, normal text is our reply. The reference section only contains those references which are newly introduced into this reply, i.e. which are not already in the manuscript.

**Reviewer 3**

*I suggest the inclusion of Hobbs (1993) within the Introduction. This is an important paper for integrating structural geology with rock mechanics (note its inclusion within Comprehensive Rock Engineering, which is probably not on the reference list for most structural geologists), focussing on fractures and joint roughness.*

We agree with the reviewer that Hobbs (1993) is an important reference to include into the Introduction section of the manuscript, in particular for its review of the fractal geometry of joint surfaces.

*I suggest also the inclusion of, for example, Weerasekara et al. (2013) and/or Cleary and Morrison (2016), in the discussion regarding grain size reduction. Further, Cleary (2001) describes an approach for direct inclusion of breakage in the Distinct Element Method, now implemented on a supercomputer. Different mechanisms of grain size reduction are noted, and their energy approach to size reduction in DEM could well be applied in consideration of the mechanisms active during evolution of a fracture surface.*

If future work was to include the simulation of grain size reduction, the "particle replacement" approach developed by Cleary (2001) could certainly be investigated as a computationally less expensive option to the full modeling of grain fracture as described by Thornton et al. (2004) or Abe and Mair (2005), although some details how this would work in conjunction with the bonded particle model, i.e. in the situation when the fracturing particle is still bonded to another particle, are not immediately clear. Also, the method appears to depend strongly on calibration data for the grain fracture process, which might not be readily available for the high stress / low strain rate conditions relevant for the triaxial deformation tests. This would probably be particularly relevant in light of the strong stress dependence of the relative importance of different comminution mechanisms (grain splitting vs. abrasion) shown by Mair and Abe (2011) under similar conditions.

We will add the following to the "Discussion" section of the manuscript after the sentence ending in line 371: "A computationally less expensive option to include grain size reduction into the numerical models might be to adapt the empirical particle replacement approach developed by Cleary (2001) to the specific requirements of the simulation of rock fracture under triaxial loading. However, as Weerasekara et al. (2013) point out, this approach is strongly dependent on the availability of good calibration data for the grain fracture under the specific stress and strain rate conditions of the process modeled."

*A figure would be helpful in 3.1, around lines 163-196, for visualisation of the model and the boundary constraints.*

The model and the boundary conditions are shown in panel (a) of Figure 1 on page 4 of the manuscript. We will add a reference to Figure 1a at line 170.

References:

Mair K. and S. Abe, 2011, Breaking Up: Comminution Mechanisms in Sheared Simulated Fault Gouge, Pure. Appl. Geophys., 168(12), 2277-2288, doi: 10.1007/s00024-011-0266-6

Thornton, C., Ciomocos, M.T. and M.J. Adams, 2004, Numerical simulations diametrical compression tests on agglomerates, Powder Technology 140, 258 – 267, doi: 10.1016/j.powtec.2004.01.022

55

---

## Referee Report (RR1)

The authors have responded to all of my concerns adequately.